# PHYSICS-INFORMED DIFFUSION MODELS

**Jan-Hendrik Bastek**
Dept. of Mechanical and Process Eng.
ETH Zurich
Zurich, Switzerland
`jbastek@ethz.ch`

**WaiChing Sun**
Dept. of Civil Eng. and Eng. Mechanics
Columbia University
New York, NY, USA
`wsun@columbia.edu`

**Dennis M. Kochmann**
Dept. of Mechanical and Process Eng.
ETH Zurich
Zurich, Switzerland
`dmk@ethz.ch`

## ABSTRACT

Generative models such as denoising diffusion models are quickly advancing their ability to approximate highly complex data distributions. They are also increasingly leveraged in scientific machine learning, where samples from the implied data distribution are expected to adhere to specific governing equations. We present a framework that unifies generative modeling and partial differential equation fulfillment by introducing a first-principle-based loss term that enforces generated samples to fulfill the underlying physical constraints. Our approach reduces the residual error by up to two orders of magnitude compared to previous work in a fluid flow case study and outperforms task-specific frameworks in relevant metrics for structural topology optimization. We also present numerical evidence that our extended training objective acts as a natural regularization mechanism against overfitting. Our framework is simple to implement and versatile in its applicability for imposing equality and inequality constraints as well as auxiliary optimization objectives.

## 1 INTRODUCTION

Denoising diffusion models (Sohl-Dickstein et al., 2015) are generative models that have gained popularity due to their success in learning intricate data distributions across various modalities, including images (Ho et al., 2020; Nichol et al., 2021; Rombach et al., 2021), videos (Ho et al., 2022a;b), graphs (Niu et al., 2020; Hoogeboom et al., 2022), text (Li et al., 2022; Wu et al., 2023), and audio waveforms (Kong et al., 2021). Originally popularized within the image generation community, their outstanding representation abilities are also increasingly leveraged in the context of *scientific machine learning*. Diffusion models have been used, among others, to upscale low-fidelity data to reduce computational costs (Shu et al., 2023), design new molecules (Xu et al., 2022) or materials (Xie et al., 2022; Düreth et al., 2023) with desired properties, or generate metamaterial unit cells tailored to a given stress-strain response in complex mechanical settings (Bastek & Kochmann, 2023). For such inverse problems, a key strength is their ability to efficiently capture the full distribution of feasible solutions and designs rather than focusing on a single deterministic outcome.

A common thread of all those applications is the explicit knowledge of the underlying governing equations, which such implied distribution must obey. Often, the training data does not stem from experimental observations but is rather generated by numerical simulations, which ensure that those equations are fulfilled. Nevertheless, diffusion models are traditionally trained on a purely *data-driven* objective (Xie et al., 2022; Buehler, 2022; Bastek & Kochmann, 2023; Vlassis & Sun, 2023; Sardar et al., 2023; Li et al., 2023; Lienen et al., 2024) and hence do not strictly enforce the intrinsic constraints during model training. Consequently, samples generated from such models may be

---

Code is available at `https://github.com/jhbastek/PhysicsInformedDiffusionModels`.

statistically aligned with the training data but may not meet the required precision in scientific applications, where adherence to the underlying physics is crucial for deployment (Jacobsen et al., 2024).

More recently, efforts have been made to ensure that samples generated from the learned distribution conform to the known constraints. Examples include imitating human motion (Yuan et al., 2022), ensuring manufacturability of proposed designs (Mazé & Ahmed, 2023), or, most importantly for physical systems, satisfying the underlying physical laws (Shu et al., 2023; Jacobsen et al., 2024), typically given by a set of partial differential equations (PDEs). Yet, a fundamental framework that rigorously addresses the incorporation of PDE constraints in such generative model settings, along with a robust mechanism to enforce these constraints akin to the well-established physics-informed neural networks (PINNs) (Raissi et al., 2019) has remained elusive. We close this gap by proposing a new framework that unifies both settings and integrates PDE constraints meaningfully into the training process. Unlike previous approaches that primarily rely on some form of post-processing of generated samples and lack a derivation from first principles, we provide this theoretical foundation and directly embed constraints into the proven representation strength of diffusion models. This approach yields state-of-the-art results in terms of PDE fulfillment. By drawing synergies between the domains of PINNs (Raissi et al., 2019) and generative modeling, we introduce *physics-informed diffusion models* (PIDMs).

**Contributions.** We make the following key contributions: **(i)** We present a novel and rigorous approach that unifies denoising diffusion models with PINNs and informs the model of PDE constraints during training, and we demonstrate via rigorous numerical experiments that our framework significantly reduces the PDE residual compared to state-of-the-art methods. **(ii)** We provide evidence that the additional training objective does *not* necessarily compromise the data likelihood; instead, we observe that it acts as an effective regularization against overfitting. **(iii)** Our approach is simple to implement into the training protocol of existing diffusion model architectures, and inference is unaffected. **(iv)** While we here focus on PDEs as a sophisticated type of equality constraint, our framework is equally applicable to other equality and inequality constraints as well as auxiliary optimization objectives (potentially also provided via a differentiable surrogate model).

## 2 BACKGROUND

### 2.1 DENOISING DIFFUSION MODELS

Denoising diffusion models are state-of-the-art generative models (Ho et al., 2020; Dhariwal & Nichol, 2021; Yang et al., 2024) that learn to gradually convert a sample of a simple prior, typically a unit Gaussian, to a sample from a generally unknown data distribution $q(\boldsymbol{x}_0)$. The idea is to introduce a fixed *forward diffusion process* that incrementally adds Gaussian noise to a given data sample $\boldsymbol{x}_0 \sim q(\boldsymbol{x}_0)$, following variance schedule $\{\beta_t \in (0,1)\}_{t=1}^T$ over $T$ steps, defined as

$$q(\boldsymbol{x}_{1:T}|\boldsymbol{x}_0) = \prod_{t=1}^T q(\boldsymbol{x}_t|\boldsymbol{x}_{t-1}), \quad q(\boldsymbol{x}_t|\boldsymbol{x}_{t-1}) = \mathcal{N}(\boldsymbol{x}_t; \sqrt{1-\beta_t}\boldsymbol{x}_{t-1}, \beta_t\boldsymbol{I}). \quad (1)$$

To generate new samples, we consider the *reverse process*,

$$q(\boldsymbol{x}_{0:T}) = p(\boldsymbol{x}_T) \prod_{t=1}^T q(\boldsymbol{x}_{t-1}|\boldsymbol{x}_t), \quad q(\boldsymbol{x}_{t-1}|\boldsymbol{x}_t) = \mathcal{N}(\boldsymbol{x}_{t-1}; \boldsymbol{\mu}(\boldsymbol{x}_t,t), \boldsymbol{\Sigma}(\boldsymbol{x}_t,t)), \quad (2)$$

in which we approximate the unknown true inverse conditional distribution $q(\boldsymbol{x}_{t-1}|\boldsymbol{x}_t)$ with a neural network $p_\theta(\boldsymbol{x}_{t-1}|\boldsymbol{x}_t)$ parameterized by $\theta$. It aims to estimate the mean $\boldsymbol{\mu}_\theta$, while we fix the covariance to $\boldsymbol{\Sigma}(\boldsymbol{x}_t,t) = \frac{1-\bar{\alpha}_{t-1}}{1-\bar{\alpha}_t}\beta_t\boldsymbol{I} = \Sigma_t\boldsymbol{I}$ with $\bar{\alpha}_t = \prod_{i=1}^t \alpha_i$, $\alpha_t = 1-\beta_t$. The network is trained by maximizing the variational lower bound of the log-likelihood, which can be simplified to several loss terms that mainly consist of KL-divergences between two Gaussians (and are hence computable in closed form) (Sohl-Dickstein et al., 2015). While the obvious choice is to estimate $\boldsymbol{\mu}_\theta$, alternative parameterizations are possible. We can obtain the mean $\boldsymbol{\mu}_t$ at timestep $t$ via a combination of the reparameterization trick and Bayes' theorem (Ho et al., 2020; Kingma & Welling, 2013):

$$\boldsymbol{\mu}_t(\boldsymbol{x}_t, \boldsymbol{\epsilon}_t) = \frac{1}{\sqrt{\alpha_t}}\left(\boldsymbol{x}_t - \frac{\beta_t}{\sqrt{1-\bar{\alpha}_t}}\boldsymbol{\epsilon}_t\right) = \frac{\sqrt{\alpha_t}(1-\bar{\alpha}_{t-1})}{1-\bar{\alpha}_t}\boldsymbol{x}_t + \frac{\sqrt{\bar{\alpha}_{t-1}}\beta_t}{1-\bar{\alpha}_t}\boldsymbol{x}_0, \quad (3)$$

where $\boldsymbol{\epsilon}_t$ represents Gaussian noise to diffuse $\boldsymbol{x}_0$ to $\boldsymbol{x}_t$. Since $\boldsymbol{x}_t$ is known during training, predicting $\boldsymbol{\mu}_t$ fixes the Gaussian noise $\boldsymbol{\epsilon}_t$ or the clean signal $\boldsymbol{x}_0$ and vice versa, and we can equivalently train the model to predict these quantities. While Ho et al. (2020) simplified the training to minimize an unweighted noise mismatch, we here consider the loss

$$L(\theta) := \mathbb{E}_{t,\boldsymbol{x}_0,\boldsymbol{\epsilon}} \left[ \lambda_t \left\| \boldsymbol{x}_0 - \hat{\boldsymbol{x}}_0 \left( \boldsymbol{x}_t(\boldsymbol{x}_0, \boldsymbol{\epsilon}), t \right) \right\|^2 \right], \tag{4}$$

where $\hat{\boldsymbol{x}}_0$ is the model *estimate* of the clean signal (omitting the parametric dependence on $\theta$ for conciseness), and $\lambda_t$ is set to Min-SNR-5 weighting (Hang et al., 2023). Note that equation 4 is equivalent to the mean squared error of the $\boldsymbol{\epsilon}_t$ or $\boldsymbol{\mu}_t$ mismatch up to a time-dependent weighting factor (Ho et al., 2020; Salimans & Ho, 2022).

## 2.2 Assembly of governing equations

Physical laws are typically formulated as a set of PDEs over a domain $\Omega \subset \mathbb{R}^d$, expressed as

$$\boldsymbol{\mathcal{F}}[\boldsymbol{u}(\boldsymbol{\xi})] = \boldsymbol{0}, \quad \boldsymbol{\xi} = (\xi_1, \xi_2, \cdots, \xi_d)^\mathsf{T} \in \Omega, \quad \boldsymbol{u} = (u_1(\boldsymbol{\xi}), u_2(\boldsymbol{\xi}), \cdots, u_c(\boldsymbol{\xi}))^\mathsf{T} \in \mathbb{R}^c, \tag{5}$$

with boundary conditions

$$\boldsymbol{\mathcal{B}}[\boldsymbol{u}(\boldsymbol{\xi})] = \boldsymbol{0}, \quad \boldsymbol{\xi} \in \partial\Omega, \tag{6}$$

where $\boldsymbol{\mathcal{F}}$ is a differential operator, $\boldsymbol{\mathcal{B}}$ is a boundary condition operator, $\partial\Omega$ is the boundary of domain $\Omega$, and $\boldsymbol{u}$ is the sought solution field that satisfies the set of PDEs for all $\boldsymbol{\xi} \in \Omega$ and boundary conditions on $\partial\Omega$. We emphasize that we reserve $\boldsymbol{\xi}$ for *spatial* coordinates defined on a physical domain, while $\boldsymbol{x}$ refers to data, which here typically contains the solution field on a discretized set of spatial coordinates, as elaborated below.

We assume that samples generated from the model $\boldsymbol{x}_0 \sim p_\theta(\boldsymbol{x}_0)$ must satisfy equation 5 and equation 6. In this study, we mainly consider image-based architectures common in diffusion models. More formally, we consider a discretized pixel grid $\Omega^h \subset \mathbb{Z} \times \mathbb{Z}$ (which may serve as an approximation of the continuous domain $\mathbb{R}^2$), where $\partial\Omega^h$ consists of the boundary pixels. For $n \times n$ pixels, the model's output is thus defined over $\boldsymbol{x}_0 \in \mathbb{R}^{c \times n \times n}$. For instance, in the context of Darcy flow problems in porous media (Jacobsen et al., 2024) $\boldsymbol{x}_0$ describes the pressure field, or in the context of solid mechanics the displacement field (Gao et al., 2022). While PINNs (Raissi et al., 2019) establish an explicit map $\mathcal{N} : \boldsymbol{\xi} \mapsto \boldsymbol{u}$ between the input and solution field, which enables the computation of required derivatives for $\boldsymbol{\mathcal{F}}$ (and potentially $\boldsymbol{\mathcal{B}}$) via automatic differentiation, we here approximate those derivatives via finite differences or comparable methods that directly operate on $\boldsymbol{x}_0$. The residual $\boldsymbol{\mathcal{R}}(\boldsymbol{x}_0)$ of the (discretized) solution field $\boldsymbol{x}_0$ is then established as a measure of the discrepancy between the generated sample $\boldsymbol{x}_0$ and the governing equations it is expected to satisfy. It is defined by the corresponding PDEs and the respective boundary conditions. More precisely, we stack both contributions into a vector as

$$\boldsymbol{\mathcal{R}}(\boldsymbol{x}_0) := \begin{bmatrix} \boldsymbol{\mathcal{F}}[\boldsymbol{x}_0] \\ \boldsymbol{\mathcal{B}}[\boldsymbol{x}_0] \end{bmatrix}. \tag{7}$$

For evaluation, we introduce the mean absolute error $\mathcal{R}_{\mathrm{MAE}}(\boldsymbol{x}_0)$ as defined in equation 28 in Appendix A.6.

## 3 Physics-informed diffusion models

We explore a scenario in which our generative diffusion model must learn a distribution whose samples are to comply with a set of governing equations, i.e., $\boldsymbol{\mathcal{R}}(\boldsymbol{x}_0) = \boldsymbol{0}$. In the usual data-driven setting, this is only indirectly ensured through the training data, typically collected from a forward simulator that produces data points which inherently follow the governing equations. These fully describe the physical system and provide us with, in principle, an *infinite* source of information (Rixner & Koutsourelakis, 2021), whereas solved instances $\{\boldsymbol{x}_0^1, \cdots, \boldsymbol{x}_0^n\}$, posing as training data, represent only a finite set of evaluations in terms of the sought solution fields.[1] Thus, our strategy is to first state our optimization objective based on the more fundamental governing equations and only subsequently incorporate training data.

---

[1] Even if we assume access to infinite training samples, the residual information may still be beneficial, since it typically imposes constraints on higher-order derivatives (comparable to Sobolev training (Czarnecki et al., 2017)).

### 3.1 CONSIDERATION OF PDE CONSTRAINTS

We maintain the probabilistic perspective of generative models and introduce the residuals as *virtual observables* (Rixner & Koutsourelakis, 2021) $\hat{r} = 0$, which we consider to be sampled from the distribution

$$q_{\mathcal{R}}(\hat{r}|x_0) = \mathcal{N}(\hat{r}; \mathcal{R}(x_0), \sigma^2 I). \tag{8}$$

In the limit $\sigma^2 \to 0$, this recovers the deterministic setting of strictly enforcing the residual equations, as all probability mass is concentrated in $\mathcal{R}(x_0)$. This can be used to compute the virtual likelihood $p_\theta(\hat{r})$, which we expand in terms of drawn samples $x_0$ as

$$p_\theta(\hat{r}) = \int p_\theta(\hat{r}, x_0) \, dx_0 = \int q_{\mathcal{R}}(\hat{r}|x_0) p_\theta(x_0) \, dx_0 = \mathbb{E}_{x_0 \sim p_\theta(x_0)} q_{\mathcal{R}}(\hat{r}|x_0). \tag{9}$$

This factorization of $p_\theta(\hat{r}, x_0)$ is reasonable as the residual follows $x_0$ via equation 8. The goal is to maximize the usual log-likelihood over the virtual observable $\hat{r}$, i.e.,

$$\arg\max_\theta \mathbb{E}_{\hat{r}} \left[ \log p_\theta(\hat{r}) \right] = \arg\max_\theta \mathbb{E}_{x_0 \sim p_\theta(x_0)} \left[ \log q_{\mathcal{R}}(\hat{r} = 0|x_0) \right]. \tag{10}$$

Thus, any samples from $p_\theta(x_0)$ are evaluated on their log-likelihood via equation 8, which can also be understood as a probabilistic reinterpretation of the loss used to train PINNs (Raissi et al., 2019). Alternatively, equation 10 can also be understood as sampling from an unnormalized target distribution assembled by evaluating $\log q_{\mathcal{R}}$ over $x_0$, as is often considered in data-free settings (Zhang & Chen, 2022; Vargas et al., 2023; Berner et al., 2024). Our setting, however, differs as we consider the more common scenario of applying diffusion models to learn a data distribution, as elaborated as follows.

### 3.2 CONSIDERATION OF OBSERVED DATA

We emphasize our focus on a *generative* model class in which training data is typically available. This is crucial for two main reasons: first, identifying solutions $x_0$ that satisfy the governing equations is non-trivial, and we may understand the obtained training data as guidance for the model towards feasible solutions, which may accelerate the training. Alternatively, the constraints may be far from a "well-posed" problem and admit a large set of solutions, while we are only interested in a small subset of those reflected by the data. Second, equation 10 contains any distribution that produces samples with zero residuals, and the model may simply collapse to a single solution instance (such as in the classical PINN setup), which is not the use case for a generative scenario. Rather, we aim to leverage the proven capabilities of generative models to learn complex distributions, such as optimal mechanical designs or fluid flows based on some conditioning, while ensuring adherence to the physical laws. Even if no data is available, we may introduce an uninformative prior such as a unit Gaussian to promote the exploration of different solutions, which we briefly investigate in Appendix A.6.1.

We thus extend the objective equation 10 by including the usual data likelihood term, which is taken as the standard optimization objective in data-driven diffusion models as

$$\arg\max_\theta \mathbb{E}_{x_0 \sim q(x_0)} \left[ \log p_\theta(x_0) \right] + \mathbb{E}_{x_0 \sim p_\theta(x_0)} \left[ \log q_{\mathcal{R}}(\hat{r} = 0|x_0) \right]. \tag{11}$$

We show in Appendix A.1 that, if diffusion models are interpreted as score-based models, a straightforward inclusion of the virtual (residual) likelihood in the loss function recovers a consistent training objective in the sense that the optimal score model also maximizes the introduced virtual (residual) likelihood and recovers samples from $q(x_0)$.

### 3.3 SIMPLIFICATION OF THE TRAINING OBJECTIVE

The joint loss equation 11 requires sampling not just from $q(x_0)$ but also $p_\theta(x_0)$, which is costly for diffusion models due to their iterative nature. We explore two ideas to mitigate this increased complexity by considering (i) a straightforward evaluation of the residual based on the readily available $\hat{x}_0$ and (ii) an accelerated sampling strategy via denoising diffusion implicit models (DDIMs) (Song et al., 2021a).

**Mean estimation.** As seen in equation 4, the diffusion model objective can be understood as minimizing a (time-weighted) mean-squared distance between the predicted and true sample, given a noisy input. Hence, $\hat{x}_0$ is an estimate of $\mathbb{E}[x_0|x_t]$ (Song et al., 2021b). Evaluating the residual on this estimate is therefore in general not fully consistent, as $\mathcal{R}(\mathbb{E}[x_0|x_t]) \neq \mathbb{E}[\mathcal{R}(x_0|x_t)]$ (also referred to as the *Jensen gap* (Gao et al., 2017)). Only at the last sampling step will this estimate align with a generated sample $x_0$ (i.e., at $t = 1$, assuming $\beta_1 = 0$) and otherwise introduce a conflicting objective between the data and residual loss with increasing $t$. To mitigate this, we propose to increase the variance of the introduced residual likelihood with increasing $t$, while enforcing tighter adherence to the constraint as $t \to 0$.

**Sample estimation.** Alternatively, we may aim to evaluate the residual of an actual sample $x_0 \sim p_\theta(x_0)$, motivated by the implied consistency shown in Appendix A.1 (though in idealized settings). This comes at the expense of increased computational complexity due to the many forward passes required in the denoising process. We hence consider (deterministic) DDIM (Song et al., 2021a) to accelerate sampling while maintaining high sample quality, with an inherent trade-off between sample quality and the number of DDIM timesteps. We consider a simple two-step sampling of $x_0$ from any $t$ and again increase the variance as $t \to T$, where timesteps are coarser, which leads to reduced sample quality (see Appendix A.4 for further details). While we focus on diffusion models, we note that single-step generation models such as consistency models (Song et al., 2023) are a promising alternative, offering direct access to a sample.

Since the remaining derivations hold true for both the mean and sample estimation, we no longer distinguish between these estimates and denote both of them with $x_0^*$ (i.e., $x_0^* := \hat{x}_0 = \mathbb{E}[x_0|x_t]$ or $x_0^* := \text{DDIM}[x_t]$), based on which the residual can efficiently be evaluated. As $x_0^*$ is obtainable at any timestep $t$, we adapt equation 11 to

$$\arg\max_\theta \mathbb{E}_{x_0 \sim q(x_0)} [\log p_\theta(x_0)] + \mathbb{E}_{x_{1:T} \sim p_\theta(x_{1:T})} [\log q_\mathcal{R}(\hat{r} = 0|x_0^*(x_{1:T}))]. \tag{12}$$

Estimates at noisier inputs may be less accurate for aforementioned reasons, thus we penalize constraint violation less as $t \to T$ by modeling $q_\mathcal{R}(\hat{r}|x_0^*(x_{1:T}))$ for simplicity based on a scaled version of the variance scheduler of the reverse process as

$$q_\mathcal{R}(\hat{r}|x_0^*(x_{1:T})) = \prod_{t=1}^{T} q_\mathcal{R}(\hat{r}|x_0^*(x_t, t)), \quad \text{where} \quad q_\mathcal{R}(\hat{r}|x_0^*(x_t, t)) = \mathcal{N}(\hat{r}; \mathcal{R}(x_0^*), \Sigma_t/c\,I), \tag{13}$$

where $\Sigma_t$ is, as introduced before, the fixed variance of the denoising process. This idea of adjusting the temperature of the target distribution shares some similarity with the *annealed noise distribution* introduced in a concurrent work by Sanokowski et al. (2024) in the context of combinatorial optimization. The scale factor $c > 0$ introduced in equation 13, which is the only hyperparameter in our setup, effectively dictates the penalty for deviating from $\mathcal{R}(x_0^*) = 0$. Figure 1 shows a graphical illustration of this process.

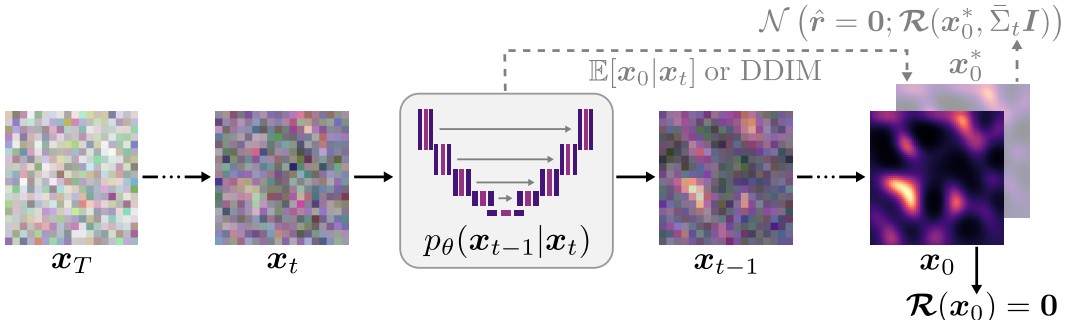

Figure 1: An approximation $x_0^*$ of the clean signal for residual evaluation can be obtained at any denoising timestep $t$ by directly considering the estimated expectation $\mathbb{E}[x_0|x_t]$ or by actual (accelerated DDIM (Song et al., 2021a)) sampling. We tighten the variance of the virtual likelihood as $t \to 0$.

As equation 12 still requires sampling over $x_{1:T} \sim p_\theta(x_{1:T})$, we simplify this by instead sampling from the available $q(x_{1:T})$, effectively ignoring the likelihood ratio (see Appendix A.3 for details).

As the original (Sohl-Dickstein et al., 2015) and physics-driven objectives are then under the same expectation, the classical data loss equation 4 can be straightforwardly extended to include a physics-informed loss, resulting in the loss of our proposed PIDM:

$$L_{\text{PIDM}}(\theta) = \mathbb{E}_{t\sim[1,T],\boldsymbol{x}_{0:T}\sim q(\boldsymbol{x}_{0:T})} \left[ \lambda_t \|\boldsymbol{x}_0 - \hat{\boldsymbol{x}}_0(\boldsymbol{x}_t, t)\|^2 + \frac{1}{2\bar{\Sigma}_t} \|\boldsymbol{\mathcal{R}}(\boldsymbol{x}_0^*(\boldsymbol{x}_t, t))\|^2 \right], \quad (14)$$

where $\bar{\Sigma}_t = \Sigma_t/c$ is the rescaled variance. Since $\Sigma_0 = 0$ for a deterministic last step, we set $\Sigma_0 \leftarrow \Sigma_1$. Algorithm 1 displays the updated training objective, requiring only minor modifications to the standard training setup (Ho et al., 2020).

---

**Algorithm 1** Physics-informed diffusion model training

---

1: Set $\bar{\Sigma}_t = \Sigma_t/c$
2: **repeat**
3:      $\boldsymbol{x}_0 \sim q(\boldsymbol{x}_0)$
4:      $t \sim \text{Uniform}\{1, \dots, T\}$
5:      $\boldsymbol{\epsilon} \sim \mathcal{N}(\boldsymbol{0}, \boldsymbol{I})$
6:      $\boldsymbol{x}_t = \sqrt{\bar{\alpha}_t}\boldsymbol{x}_0 + \sqrt{1 - \bar{\alpha}_t}\boldsymbol{\epsilon}$
7:      Estimate $\boldsymbol{x}_0^*$ via $\hat{\boldsymbol{x}}_0 = \mathbb{E}[\boldsymbol{x}_0|\boldsymbol{x}_t]$ or DDIM sampling (Song et al., 2021a)
8:      Take gradient descent step on $\nabla_\theta \left[ \lambda_t \|\boldsymbol{x}_0 - \hat{\boldsymbol{x}}_0(\boldsymbol{x}_t, t)\|^2 + \frac{1}{2\bar{\Sigma}_t} \|\boldsymbol{\mathcal{R}}(\boldsymbol{x}_0^*(\boldsymbol{x}_t, t))\|^2 \right]$
9: **until** converged

---

We point out that the model can also be trained to match the mean or noise as in Ho et al. (2020), from which we can equally assemble $\mathbb{E}[\boldsymbol{x}_0|\boldsymbol{x}_t]$ if considering mean estimation via equation 3 (see Appendix A.2 for estimation with score models) but we empirically found that this leads to larger residual errors. Also, we emphasize that PDEs can be understood as a specific instance of equality constraints and we could generally consider any differentiable forward surrogate model $\mathcal{R}_\theta(\boldsymbol{x}_0)$ that estimates some property or classification of the predicted samples to be matched, i.e., $\mathcal{R}_\theta(\boldsymbol{x}_0) = \hat{r}_{\text{target}}$. For a discussion on how to incorporate inequality constraints and auxiliary optimization objectives see Appendix A.5. Lastly, we note that the optimal scaling $c$ generally depends on the considered scenario, and it is here estimated by a simple parameter sweep. As usual in multi-objective optimization, trade-offs between the different loss contributions are expected, and $c$ must be selected so that the model is meaningfully informed by the residual loss but does not ignore the data likelihood.

## 4 EXPERIMENTS

We here present two benchmarks for distributions from which samples are implied to adhere to specific governing PDEs and constraints, as recently studied in contemporary work (Jacobsen et al., 2024; Giannone et al., 2023). To obtain intuition for the effects of the proposed loss, see the toy problem presented in Appendix A.6.1 as an instructive example.

### 4.1 DARCY FLOW

**Setup.** We first study the 2D Darcy flow equations, which describe the steady-state solution for fluid flow through a porous medium, here on a square domain $\Omega = [0,1]^2$. Generally, we follow the setup of previous studies (Jacobsen et al., 2024; Zhu & Zabaras, 2018) by sampling permeability fields $K(\boldsymbol{\xi})$ from a Gaussian random field, which are solved for their (unique) pressure distribution $p(\boldsymbol{\xi})$. Similarly, we create a training and a validation dataset of 10,000 and 1,000 datapoints, respectively, by solving the governing equations (see equation 29) for a sampled permeability field on a $64 \times 64$ grid. Second-order central finite differences are used to assemble and solve a linear system (see Jacobsen et al. (2024)), giving pairs $(\boldsymbol{K}, \boldsymbol{p})$ with $\boldsymbol{K}, \boldsymbol{p} \in \mathbb{R}^{n \times n}$. We consider a U-Net (Ronneberger et al., 2015) architecture with $64 \times 64$ pixels as in- and output dimensions that align with the considered grid and allow for the same residual evaluation via finite differences also used to create the dataset. We increase the variance of the virtual residual likelihood by setting $c = 10^{-3}$ for the mean estimation and $c = 10^{-5}$ for the sample estimation. Throughout the experiments, we observed that the mean estimation performs best with a variance of the residual likelihood around two orders of magnitude smaller than the best performance for the sample estimation. Further details of the considered setup and implementation are given in Appendices A.6.2 and A.7.

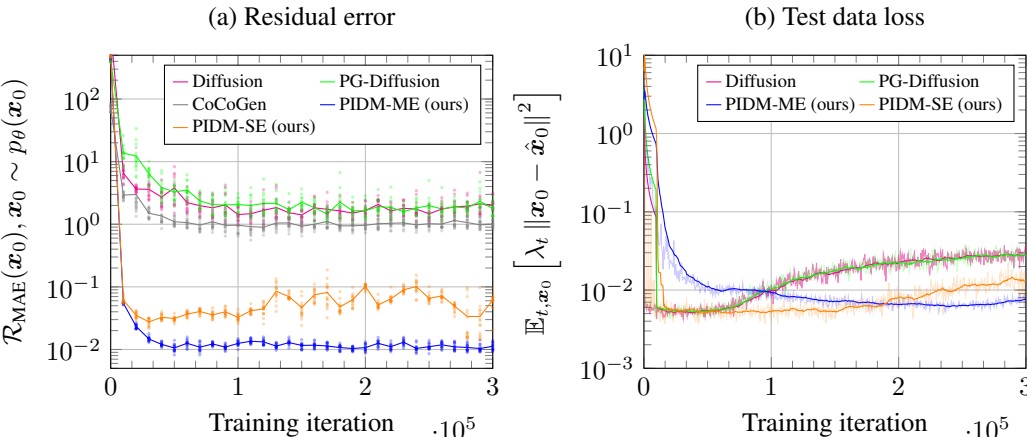

Figure 2: Evaluation of the residual error (a) and test data loss (b) during training. In (a), we generate 16 samples every 10k training iterations and plot the average (solid lines) and individual (dots) residual errors for the standard diffusion model ('Diffusion'), the physics-guided model ('PG-Diffusion') (Shu et al., 2023), CoCoGen (Jacobsen et al., 2024), and the proposed PIDM using either mean or sample estimation ('PIDM-ME' and 'PIDM-SE', respectively). Note that for CoCoGen not all samples converged, so that we excluded the non-converged data from the indicated average. In (b), we plot the data loss evaluated on a test set for the proposed PIDM variants and those frameworks that differ from ours during training.

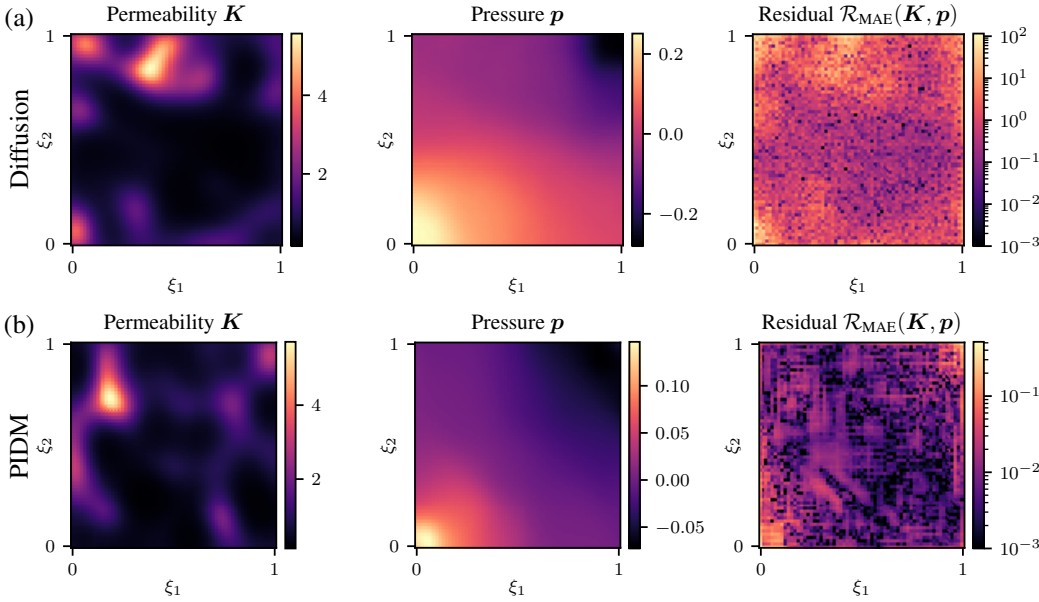

Figure 3: Generated permeability and pressure fields as well as the corresponding residual error from diffusion models trained on the Darcy flow dataset, where (a) is sampled from a standard diffusion model and (b) from our proposed PIDM with mean estimation. Additional samples are shown in Appendix A.8.1.

**Results.** To evaluate the performance of our proposed PIDM, we benchmark it against three relevant setups: (i) a model trained on the standard (purely data-driven) objective equation 4, (ii) a "physics-guided" model trained on the standard objective but using residual information as guidance, as proposed by Shu et al. (2023), and (iii) a model similar to (i) but with first-order residual corrections during inference, as described in CoCoGen (Jacobsen et al., 2024) (see Appendix A.6.2 for further

details). We examine the performance of the different diffusion model variants by tracking the evolution of the residual error of generated samples from the learned distributions $x_0 \sim p_\theta(x_0)$ alongside the test data loss throughout the training process in Figure 2. The PIDM showcases a remarkable improvement, reducing the residual error by around two orders of magnitude in comparison to the standard diffusion model. This holds for both the mean and sample (DDIM) estimation, where the former outperforms the latter variant in terms of the residual error. The mean estimation takes around 23% longer to train compared to the standard diffusion model due to the residual evaluation, whereas the sample estimation took around 69% longer due to the additional forward pass (see Appendix A.4). We could not observe a significant reduction of the residual error for the physics-guided model (Shu et al., 2023). We hypothesize that the mere inclusion of gradient information into the model seems insufficient because it does not truly enforce residual minimization. CoCoGen (Jacobsen et al., 2024) shows a moderate improvement but does not bring the residual close to values where the samples could be considered physically consistent. We display generated samples in Figure 3 and Appendix A.8.1 for both the standard diffusion model and PIDM, confirming a drastically reduced residual error over the whole domain. Figures 2b, 3b and 8 confirm that the PIDMs maintain the generative diversity with both estimation techniques and do not collapse to a single solution instance. It successfully generates permeability (and pressure) fields that adhere to the data distribution (besides respecting the PDE).

For any multi-objective loss as given in equation 14, a trade-off between the data and residual minimization is generally expected, depending on their relative weighting dictated by $c$ (Wang et al., 2021). Interestingly, Figure 2b shows that the PIDM with mean estimation eventually recovers a similar test data loss as the vanilla diffusion model and is significantly less prone to overfitting. Even more notable is that the PIDM with sample estimation initially has a similar test data loss trajectory as the vanilla model—despite the additional loss term—and also remains more robust to overfitting. This is strong evidence of the consistent data and physics loss (see Appendix A.1). Thus, sample estimation drastically improves the alignment of the two losses with only a single additional forward pass. The increased robustness to overfitting in both mean and sample estimation is a clear indicator that the model learns a more robust internal representation of the data distribution, as it is forced to generate samples that adhere to the true distribution or, equivalently, the underlying data generation mechanism—viz. the known constraints that govern the system. Thus, adding the physical (residual) loss may benefit the generative performance by enhancing its generalization capability.

## 4.2 Topology optimization

**Setup.** We consider 2D structural topology optimization as a second example. In this setting, the goal is to find the optimal material distribution $\rho_{\text{opt}}(\xi)$ that maximizes the mechanical stiffness, or equivalently, minimizes the compliance of a structure under a set of constraints that typically consist of mechanical equilibrium, boundary conditions, and a volume constraint. This is classically solved via the SIMP method based on a finite element (FE) discretization (Bendsøe & Sigmund, 2004). We again consider a square domain $\Omega = [0, 1]^2$ and benchmark our proposed PIDM to state-of-the-art frameworks (Mazé & Ahmed, 2023; Giannone et al., 2023) that also provide a dataset consisting of 30,000 optimized structures with various boundary conditions and volume constraints and two proposed test scenarios with in- and out-of-distribution boundary conditions. We train a U-Net architecture similar to the one in Section 4.1 but with a larger latent dimension and additional in- and output channels on this dataset. To ensure consistent evaluation of the residuals, we do *not* use finite differences but interpret the pixels as nodes of the underlying FE mesh to assemble a consistent stiffness matrix equivalent to the one used to generate the data. We scale the variance of the residual likelihood with $c = 0.01$ and introduce the volume constraint as an additional equality constraint with $c = 0.1$ (as the optimal topology will contain the maximum allowed material). We also introduce a slight bias to minimize compliance (setting $\lambda = 10^{-6}$ for the optimization objective, see Appendix A.5.2). Further details of the setup, model architecture, residual evaluation, and implementation are presented in Appendices A.6.3 and A.7.

**Results.** We evaluate the performance of our proposed PIDM in comparison to the standard diffusion model, PG-Diffusion (Shu et al., 2023) and CoCoGen (Jacobsen et al., 2024), as well as two recent variants specifically tailored to topology optimization. These propose to modify the sampling process by either using additional guidance models to reduce compliance and improve manufacturability (Mazé & Ahmed, 2023) or enforcing the denoising trajectory to be closer to an iterative optimization

trajectory (Giannone et al., 2023). Both methods require auxiliary datasets which complicate model training—in contrast, our method is a much simpler and well-motivated extension of the standard training without requiring any additional data or models. While training of the 'main' diffusion model takes longer for the PIDM due to the additional computational complexity and memory requirements, we note that inference is identical to the standard diffusion model and does not require additional surrogate models as, e.g., in Mazé & Ahmed (2023). Results are shown in Table 1 for the PIDM with sample estimation, which we observed to outperform the mean estimation slightly. We evaluate relevant performance metrics on the two test sets (with seen and unseen boundary conditions, respectively), with generated samples presented in Figure 4 and Appendix A.8.2.

Table 1: Performance comparison of diffusion model variants for topology optimization. We consider in- and out-of-distribution boundary conditions as described in Mazé & Ahmed (2023) and provide the median $\mathcal{R}_{\mathrm{MAE}}$ of the predicted solution fields (where applicable), the median compliance error (MDN % CE) and the mean volume fraction error (% VFE). *Giannone et al. (2023) further improve their model by running additional SIMP post-processing steps, but for a consistent comparison, we consider unprocessed samples from the model.

| Model | Size | in-distribution | | | out-of-distribution | | |
|---|---|---|---|---|---|---|---|
| | | $\mathcal{R}_{\mathrm{MAE}} \downarrow$ | MDN % CE ↓ | % VFE ↓ | $\mathcal{R}_{\mathrm{MAE}} \downarrow$ | MDN % CE ↓ | % VFE ↓ |
| Diffusion | 136M | 1.86e−3 | **−0.2** | 2.93 | 1.97e−3 | **0.3** | 2.80 |
| PG-Diffusion (Shu et al., 2023) | 136M | 1.82e−3 | 0.09 | 3.59 | 1.92e−3 | 0.81 | 3.23 |
| CoCoGen (Jacobsen et al., 2024) | 136M | 1.51e−3 | 0.14 | 4.00 | 1.56e−3 | 0.58 | 3.64 |
| TopoDiff-G (Mazé & Ahmed, 2023) | 239M | - | 0.83 | **1.49** | - | 1.82 | 1.80 |
| DOM* (Giannone et al., 2023) | 121M | - | 0.74 | 1.52 | - | 3.47 | **1.59** |
| PIDM (ours) | 136M | **1.24e−3** | 0.06 | 2.25 | **1.29e−3** | 0.56 | 1.91 |

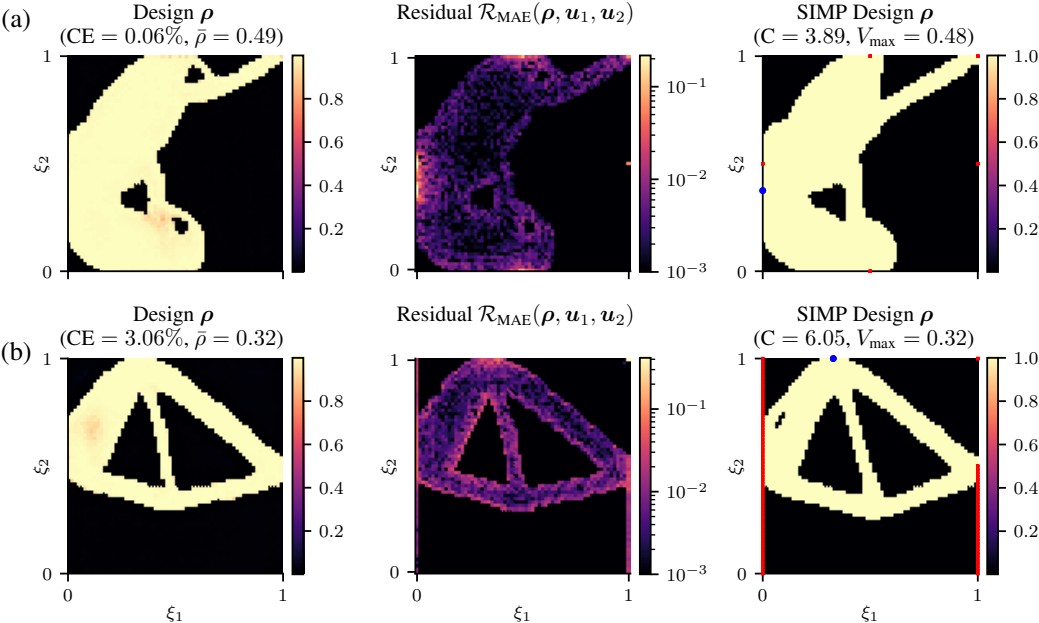

Figure 4: Generated designs, including the compliance error CE and volume $\bar{\rho}$, and the residual error (based on the displacement fields, not shown) from diffusion models trained on the SIMP dataset and the corresponding SIMP design, including the compliance C and volume $V_{\mathrm{max}}$. We plot a sample (a) from a standard diffusion model and (b) from our proposed PIDM. All samples are conditioned on the out-of-distribution test set. In the SIMP design, we indicate the applied load by a blue dot, and the given boundary conditions in red.

Importantly, the PIDM provides not only the optimized designs but also the displacement fields, which are of significant interest for mechanical analysis, e.g., to estimate the stress distribution in the

structure. We observe a residual reduced by 33% and 35% compared to the standard diffusion model on the in- and out-of-distribution set, respectively, highlighting closer adherence to the mechanical equilibrium. Somewhat surprisingly, we observe that the standard model performs very well in terms of the optimization objective, but also emphasize that it has an increased volume fraction error—thus results are not fully comparable, as more volume will allow for stiffer structures. Contrary, the PIDM only has a slight increase in the volume fraction error but significantly outperforms previous frameworks (Mazé & Ahmed, 2023; Giannone et al., 2023) in terms of compliance minimization, despite the absence of auxiliary data and surrogate models. Lastly, adaptations of PG-Diffusion (Shu et al., 2023) and CoCoGen (Jacobsen et al., 2024) also cannot match the PIDM in terms of the residual (on both test sets), though CoCoGen can reduce it slightly. Additionally, the median compliance and especially the volume fraction error perform significantly worse.

## 5 RELATED WORK

Conceptually closest to our work are two recent contributions by Shu et al. (2023) and Jacobsen et al. (2024), to which we compare the proposed PIDM extensively in Section 4. Our studies indicate that the PIDM significantly reduces the residual error of generated samples compared to both variants, especially for the Darcy flow study. Focusing on topology optimization, two recent contributions (Mazé & Ahmed, 2023; Giannone et al., 2023) propose improvements that require auxiliary data and/or surrogate models, while we here show that the PIDM, solely by being *physics-informed*, significantly outperforms both methods in terms of the optimal stiffness. In a broader context, Wang et al. (2023) optimized the conditioning variable to create an online dataset of soft robot designs, which includes physical performance and may generate samples with improved physical utility. Yet, the focus was on leveraging pre-trained models to create diverse 3D shapes. Yuan et al. (2022) leveraged diffusion models for human motion synthesis, where inference is altered by projecting the intermediate steps to a physically plausible motion that is verified via a reinforcement learning approach. Similarly, Christopher et al. (2024) projected intermediate steps to the closest point in a feasible set, which is generally unknown for more complicated constraints such as PDEs. In general, such post-processing methods may indeed mitigate some of the mismatches of the generated samples (or fulfill them exactly if the constraints are sufficiently simple). Yet, they are fundamentally limited, as they do not address the underlying distribution learned by the model.

## 6 CONCLUSION

We have unified the data-driven perspective of diffusion models with a physics-informed paradigm, enabling the models to internalize the constraints that generated samples must adhere to. Our framework significantly outperforms purely data-driven models and prior work, as verified by two highly relevant case studies, and numerical evidence hints that the PIDM obtains a more robust representation of the data distribution that is less prone to overfitting. We hope this work stimulates others to extend their generative model training objective when, besides data, further information—be it in the form of PDEs or other constraints—on the generated samples is available, as is often the case within the realm of scientific machine learning. Future work may explore more sophisticated virtual likelihood variance schedulers, which we here simply coupled to the standard denoising process with a scaling estimated by a parameter sweep. More generally, architectures with a consistent residual evaluation that do not operate on a fixed regular grid should be considered, currently restricting the studies to geometrically simple domains. Remedy can be found in graph-based architectures that can handle arbitrary meshes (Gao et al., 2022) or by employing implicit encodings of coordinates (Liu et al., 2023). Additionally, coordinate-based representations allow for exact calculation of the gradients required to evaluate the imposed PDEs via automatic differentiation, though at increased computational cost.

## ACKNOWLEDGMENTS

The authors thank François Mazé for providing the code to generate the additional datasets for the topology optimization case study and Matheus Inguaggiato Nora Rosa for the helpful discussions about the corresponding implementation of the residual evaluation.

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

# A APPENDIX

## A.1 CONSISTENCY

We here give a theoretical argument for the fact that, in an ideal setting, diffusion models trained via score-matching with an additional virtual (residual) likelihood term continue to recover the true distribution. Adopting the perspective of score-based models (Song et al., 2021b), we consider a continuous time variable $t \in [0, T]$ instead of the discrete setting. The forward diffusion process of some data $\boldsymbol{x} \in \mathbb{R}^D$ can then generally be described by the stochastic differential equation (SDE)

$$\mathrm{d}\boldsymbol{x} = \boldsymbol{F}_t \boldsymbol{x} \, \mathrm{d}t + \boldsymbol{G}_t \, \mathrm{d}\boldsymbol{w}, \tag{15}$$

where $\boldsymbol{w}$ is a Wiener process, and the drift and diffusion coefficients $\boldsymbol{F}_t \in \mathbb{R}^{D \times D}$ and $\boldsymbol{G}_t \in \mathbb{R}^{D \times D}$, respectively, are chosen so that the transition kernel is a simple Gaussian that can be computed in closed form. Here, the popular diffusion variants proposed by Ho et al. (2020) and DDIM (Song et al., 2021b) correspond to their continuous counterparts $\boldsymbol{F}_t = \frac{1}{2} \frac{\mathrm{d} \log \alpha_t}{\mathrm{d}t} \boldsymbol{I}$ and $\boldsymbol{G}_t = \sqrt{-\frac{\mathrm{d} \log \alpha_t}{\mathrm{d}t}} \boldsymbol{I}$, where $\alpha_t$ is continuously decreasing from $\alpha_0 \approx 1$ to $\alpha_T \approx 0$. There is no unique reverse diffusion process, but common choices of the corresponding reverse SDE are included in the parameterization proposed by Zhang & Chen (2023),

$$\mathrm{d}\boldsymbol{x} = \left[ \boldsymbol{F}_t \boldsymbol{x} - \frac{1 + \lambda^2}{2} \boldsymbol{G}_t \boldsymbol{G}_t^T \nabla \log p_t(\boldsymbol{x}) \right] \mathrm{d}\bar{t} + \lambda \boldsymbol{G}_t \, \mathrm{d}\bar{\boldsymbol{w}}, \tag{16}$$

where $\bar{\boldsymbol{w}}$ is a Wiener process that runs, as $\mathrm{d}\bar{t}$, backwards in time, and $\lambda \geq 0$. If the score $\nabla \log p_t(\boldsymbol{x})$ is known, we can generate new samples by sampling from the known prior distribution $p(\boldsymbol{x}_T)$ and applying equation 16. DDPM (Ho et al., 2020) and (deterministic) DDIM (Song et al., 2021a) then correspond to certain parameterizations ($\lambda = 1$ and $\lambda = 0$, respectively) and discretizations of equation 16 (Song et al., 2021b; Zhang & Chen, 2023). We can show that a straightforward extension of the usual score-matching objective with the virtual residual likelihood is consistent as it continues to recover the data distribution:

**Proposition 1.** *(Consistency) Let $p(\boldsymbol{x}_0)$ be a distribution with samples $\boldsymbol{x}_0 \sim p(\boldsymbol{x}_0)$ satisfying some constraint $\mathcal{R}(\boldsymbol{x}_0) = \boldsymbol{0}$. Consider*

$$\boldsymbol{s}_{\mathrm{opt}} = \arg \min_{\boldsymbol{s}} \mathbb{E}_{t \sim \mathrm{Unif}[0,T]} \mathbb{E}_{p(\boldsymbol{x}_0) p(\boldsymbol{x}_t | \boldsymbol{x}_0)} \left[ \Lambda(t) \left\| \nabla \log p(\boldsymbol{x}_t | \boldsymbol{x}_0) - \boldsymbol{s}(\boldsymbol{x}_t, t) \right\|^2 - \log q_{\mathcal{R}}(\hat{\boldsymbol{r}} | \boldsymbol{x}_0^*(\boldsymbol{x}_t)) \right], \tag{17}$$

*where $\Lambda(t) > 0$ is a time-dependent weight and $\boldsymbol{x}_0^*$ is obtained by solving the reverse SDE equation 16 initiated at $\boldsymbol{x}_t$ with score $\boldsymbol{s}(\boldsymbol{x}_t, t)$. Then, solving the reverse SDE equation 16 from $\boldsymbol{x}_T \sim p(\boldsymbol{x}_T)$ with $\boldsymbol{s}_{\mathrm{opt}}$ as the score for $\boldsymbol{x}_0$ corresponds to sampling from $p(\boldsymbol{x}_0)$.*

*Proof.* It is well-known that score matching $\nabla \log p(\boldsymbol{x}_t | \boldsymbol{x}_0)$ is equivalent to matching $\nabla \log p(\boldsymbol{x}_t)$ (Vincent, 2011). Assuming perfect recovery of the score, i.e., $\boldsymbol{s}(\boldsymbol{x}, t) = \nabla \log p(\boldsymbol{x}_t)$ for all $\boldsymbol{x}_t, t$, the marginal distribution $p^*(\boldsymbol{x}_t)$ of equation 16 matches the forward diffusion $p(\boldsymbol{x}_t)$ for all $0 \leq t \leq T$ independent of $\lambda$ when $p(\boldsymbol{x}_T) = p^*(\boldsymbol{x}_T)$ (Zhang & Chen, 2023, Prop. 1). To generate new samples, we may start from any latent $\boldsymbol{x}_t^* \sim p^*(\boldsymbol{x}_t)$, which can hence equivalently be obtained via the forward marginal $\boldsymbol{x}_t \sim p(\boldsymbol{x}_t)$, and solve equation 16 for $\boldsymbol{x}_0^* \sim p(\boldsymbol{x}_0)$. As we consider virtual observables $\hat{\boldsymbol{r}} = \boldsymbol{0}$ introduced via $q_{\mathcal{R}}(\hat{\boldsymbol{r}} | \boldsymbol{x}_0^*) = \mathcal{N}(\hat{\boldsymbol{r}}; \mathcal{R}(\boldsymbol{x}_0^*), \sigma^2 \boldsymbol{I})$ we have

$$-\log q_{\mathcal{R}}(\hat{\boldsymbol{r}} | \boldsymbol{x}_0^*) = \frac{1}{2} \| \mathcal{R}(\boldsymbol{x}_0^*) / \sigma^2 \|^2 + C, \tag{18}$$

where $C$ is a constant that does not depend on $\boldsymbol{x}_0^*$ (and hence $\boldsymbol{s}$). Since $\boldsymbol{x}_0^* \sim p(\boldsymbol{x}_0)$ satisfies $\mathcal{R}(\boldsymbol{x}_0^*) = \boldsymbol{0}$ by assumption, the optimal score model is also a minimizer of the (negative) virtual log-likelihood equation 18 and $\boldsymbol{s}_{\mathrm{opt}}$ recovers the optimal score. Hence, solving equation 16 from $\boldsymbol{x}_T \sim p(\boldsymbol{x}_T)$ for $\boldsymbol{x}_0$ with $\boldsymbol{s}_{\mathrm{opt}}$ as the score generates samples from the true distribution $\boldsymbol{x}_0 \sim p(\boldsymbol{x}_0)$. □

*Remark.* In practice, equation 16 has to be discretized, since no closed-form solution is available. Hence, even for a perfect score model, this numerical approximation will introduce a bias (Zhang & Chen, 2023). Also, note that we are free to choose DDPM (Ho et al., 2020) or DDIM (Song et al., 2021a) to discretize equation 16 since the optimal score is unaffected by this choice.

Lastly, we emphasize that minimizing the residual error on some intermediate latent variables introduces inconsistencies. Assume some general forward conditional marginal of the form $p(\boldsymbol{x}_t|\boldsymbol{x}_0) = \mathcal{N}(\boldsymbol{x}_t; \alpha_t\boldsymbol{x}_0, \sigma_t^2\boldsymbol{I})$, from which we can sample via $\boldsymbol{x}_t = \alpha_t\boldsymbol{x}_0 + \sigma_t^2\boldsymbol{\epsilon}$, where $\boldsymbol{\epsilon} \sim \mathcal{N}(\boldsymbol{0}, \boldsymbol{I})$. But in general, $\mathcal{R}(\alpha_t\boldsymbol{x}_0 + \sigma_t^2\boldsymbol{\epsilon}) \neq \boldsymbol{0}$; thus enforcing $\mathcal{R}(\boldsymbol{x}_t) = \boldsymbol{0}$ does not align with the underlying marginal distribution.

## A.2 Mean estimation for score-based models

As summarized in Appendix A.1, score-based models are an alternative perspective on diffusion models that may be understood as their continuous-time generalization (Song et al., 2021b). Here, the model learns the data score $\boldsymbol{s}_\theta(\boldsymbol{x}_t, t) \approx \nabla \log p_t(\boldsymbol{x})$, with which we can solve the reverse SDE equation 16 to generate new samples. For a forward conditional marginal of form $p(\boldsymbol{x}_t|\boldsymbol{x}_0) = \mathcal{N}(\boldsymbol{x}_t; \alpha_t\boldsymbol{x}_0, \sigma_t^2\boldsymbol{I})$, we can obtain $\hat{\boldsymbol{x}}_0 = \mathbb{E}[\boldsymbol{x}_0|\boldsymbol{x}_t]$ as (see e.g., Kingma et al. (2021))

$$\hat{\boldsymbol{x}}_0 = \frac{1}{\alpha_t} \left( \boldsymbol{x}_t + \sigma_t^2 \boldsymbol{s}_\theta(\boldsymbol{x}_t, t) \right), \tag{19}$$

based on which the virtual residual likelihood can be estimated.

## A.3 Details on the simplified training objective

As shown in Section 3, we arrive at an optimization objective equation 12 that requires sampling over latents from the learned distribution $\boldsymbol{x}_{1:T} \sim p_\theta(\boldsymbol{x}_{1:T})$. We simplify this by instead sampling from the available $q(\boldsymbol{x}_{1:T})$, which can be understood as ignoring the likelihood ratio, as we aim to minimize

$$\mathbb{E}_{\boldsymbol{x}_0 \sim q(\boldsymbol{x}_0)} \left[ -\log p_\theta(\boldsymbol{x}_0) \right] + \mathbb{E}_{\boldsymbol{x}_{1:T} \sim p_\theta(\boldsymbol{x}_{1:T})} \left[ -\log q_{\mathcal{R}}(\hat{\boldsymbol{r}} = \boldsymbol{0}|\boldsymbol{x}_0^*(\boldsymbol{x}_{1:T})) \right]$$

$$\leq \mathbb{E}_{q(\boldsymbol{x}_{0:T})} \left[ \log \frac{q(\boldsymbol{x}_{1:T}|\boldsymbol{x}_0)}{p_\theta(\boldsymbol{x}_{0:T})} \right] + \mathbb{E}_{\boldsymbol{x}_{1:T} \sim p_\theta(\boldsymbol{x}_{1:T})} \left[ -\log q_{\mathcal{R}}(\hat{\boldsymbol{r}} = \boldsymbol{0}|\boldsymbol{x}_0^*(\boldsymbol{x}_{1:T})) \right]$$

$$= \mathbb{E}_{q(\boldsymbol{x}_{0:T})} \left[ \log \frac{q(\boldsymbol{x}_{1:T}|\boldsymbol{x}_0)}{p_\theta(\boldsymbol{x}_{0:T})} \right] + \mathbb{E}_{\boldsymbol{x}_{1:T} \sim q(\boldsymbol{x}_{1:T})} \left[ -\frac{p_\theta(\boldsymbol{x}_{1:T})}{q(\boldsymbol{x}_{1:T})} \log q_{\mathcal{R}}(\hat{\boldsymbol{r}} = \boldsymbol{0}|\boldsymbol{x}_0^*(\boldsymbol{x}_{1:T})) \right]$$

$$\approx \mathbb{E}_{q(\boldsymbol{x}_{0:T})} \left[ \log \frac{q(\boldsymbol{x}_{1:T}|\boldsymbol{x}_0)}{p_\theta(\boldsymbol{x}_{0:T})} \right] + \mathbb{E}_{\boldsymbol{x}_{1:T} \sim q(\boldsymbol{x}_{1:T})} \left[ -\log q_{\mathcal{R}}(\hat{\boldsymbol{r}} = \boldsymbol{0}|\boldsymbol{x}_0^*(\boldsymbol{x}_{1:T})) \right]$$

$$= \mathbb{E}_{q(\boldsymbol{x}_{0:T})} \left[ \log \frac{q(\boldsymbol{x}_{1:T}|\boldsymbol{x}_0)}{p_\theta(\boldsymbol{x}_{0:T})} - \log q_{\mathcal{R}}(\hat{\boldsymbol{r}} = \boldsymbol{0}|\boldsymbol{x}_0^*(\boldsymbol{x}_{1:T})) \right].$$

We justify this simplification by the assumption that optimizing the variational bound will bring $p_\theta(\boldsymbol{x}_{1:T})$ close to $q(\boldsymbol{x}_{1:T})$ and thus reduce this bias with ongoing model training. Evaluating the final expression gives us the presented physics-informed diffusion model loss equation 14.

## A.4 DDIM sampling

For completeness, we here provide the deterministic DDIM sampling scheme derived by Song et al. (2021a) in terms of $\hat{\boldsymbol{x}}_0$:

$$\boldsymbol{x}_{\tau_{i-1}} = \sqrt{\bar{\alpha}_{\tau_{i-1}}}\hat{\boldsymbol{x}}_0(\boldsymbol{x}_{\tau_i}, t) + \sqrt{\frac{1 - \bar{\alpha}_{\tau_{i-1}}}{1 - \bar{\alpha}_{\tau_i}}} \cdot \left( \boldsymbol{x}_{\tau_i} - \sqrt{\bar{\alpha}_{\tau_i}}\hat{\boldsymbol{x}}_0(\boldsymbol{x}_{\tau_i}, t) \right). \tag{20}$$

Generally, $\tau$ is a sub-sequence of the full denoising sequence $[1, \dots, T]$. During training, $\boldsymbol{x}_t$ is available and we aim to estimate $\boldsymbol{x}_0$ in a given number of reduced timesteps, considering a sequence $\tau = [1, \dots, t]$. As we empirically found no notable improvement by providing multiple intermediate timesteps, but this comes with the cost of the additional forward passes, we set $\tau = [1, t]$. Hence, we sample $\boldsymbol{x}_1$ from $\boldsymbol{x}_t$ in one step via equation 20 and then obtain $\boldsymbol{x}_0$ by $\hat{\boldsymbol{x}}_0(\boldsymbol{x}_1, t = 1)$, resulting in two forward passes of the model. Note that we reduce the impact of worse estimates at noisier timesteps by the increased variance of the virtual residual likelihood, penalizing deviations from $\mathcal{R}(\boldsymbol{x}_0) = \boldsymbol{0}$ less.

## A.5 OTHER CONSTRAINT TYPES AND AUXILIARY OPTIMIZATION OBJECTIVES

### A.5.1 INEQUALITY CONSTRAINTS

Inequality constraints are also common in physics (e.g., the second law of thermodynamics). Consideration of these constraints of type

$$h(\boldsymbol{x}_0) \leq h_{\max} \tag{21}$$

can be introduced via $\mathcal{R}_{\mathrm{ineq}} = \mathrm{ReLU}(h(\boldsymbol{x}_0) - h_{\max}) = \max(0, h(\boldsymbol{x}_0) - h_{\max})$, which we analogously introduce into the variational loss as the Gaussian $q_{\mathcal{R}_{\mathrm{ineq}}}(\hat{r}_{\mathrm{ineq}}|\boldsymbol{x}_0^*(\boldsymbol{x}_t, t)) = \mathcal{N}(\hat{r}_{\mathrm{ineq}}; \mathcal{R}_{\mathrm{ineq}}(\boldsymbol{x}_0^*), \Sigma_t/c)$ with $\hat{r}_{\mathrm{ineq}} = 0$.

### A.5.2 OPTIMIZATION OBJECTIVES

Optimization objectives of the form

$$\min \mathcal{J}(\boldsymbol{x}_0) \tag{22}$$

can also be considered by treating the optimum as a pseudo-observable $\hat{j}_{\mathrm{opt}}$ and a similar strategy as before. In general, however, this optimum is—unlike $\hat{r}$ and $\hat{r}_{\mathrm{ineq}}$—typically unknown. As a remedy, we may extend the mismatch of the actual and optimal objective by a pseudo-observable $\hat{r}_{\mathrm{opt}} = 0$ and pose this as a sample from the exponential distribution, as shown by Rixner & Koutsourelakis (2021):

$$\mathcal{J}(\boldsymbol{x}_0) - \hat{j}_{\mathrm{opt}} + \hat{r}_{\mathrm{opt}} \sim \mathrm{Expon}(\lambda), \tag{23}$$

where

$$\mathrm{Expon}(x; \lambda) = \begin{cases} \lambda e^{-\lambda x} & x \geq 0 \\ 0 & x < 0 \end{cases}. \tag{24}$$

By similar reasoning as before, we introduce

$$q_{\mathcal{J}}(\hat{r}_{\mathrm{opt}}|\boldsymbol{x}_0^*(\boldsymbol{x}_{1:T})) = \prod_{t=1}^{T} q_{\mathcal{J}}(\hat{r}_{\mathrm{opt}}|\boldsymbol{x}_0^*(\boldsymbol{x}_t, t)),$$

$$\text{where } q_{\mathcal{J}}(\hat{r}_{\mathrm{opt}}|\boldsymbol{x}_0^*(\boldsymbol{x}_t, t)) = \lambda e^{-\lambda(\mathcal{J}(\boldsymbol{x}_0^*) - \hat{j}_{\mathrm{opt}})}, \tag{25}$$

and observe that the log-likelihood

$$\log q_{\mathcal{J}}(\hat{r}_{\mathrm{opt}}|\boldsymbol{x}_0^*(\boldsymbol{x}_{1:T})) = \sum_{t=1}^{T} \left[ \log(\lambda) - \lambda \mathcal{J}(\boldsymbol{x}_0^*(\boldsymbol{x}_t, t)) + \lambda \hat{j}_{\mathrm{opt}} \right] \tag{26}$$

decouples $\hat{j}_{\mathrm{opt}}$ from $\boldsymbol{x}_0^*$ due to the properties of the exponential distribution. Therefore, knowledge of $\hat{j}_{\mathrm{opt}}$ is not required for training. The loss function is hence extended to

$$L_{\mathrm{PIDM\text{-}opt}}(\theta) = \mathbb{E}_{t,\boldsymbol{x}_{0:T}} \left[ \lambda_t \|\boldsymbol{x}_0 - \hat{\boldsymbol{x}}_0(\boldsymbol{x}_t, t)\|^2 + \frac{1}{2\bar{\Sigma}_t} \|\mathcal{R}(\boldsymbol{x}_0^*(\boldsymbol{x}_t, t))\|^2 + \lambda \mathcal{J}(\boldsymbol{x}_0^*(\boldsymbol{x}_t, t)) \right]. \tag{27}$$

Note that we could also couple the parameter $\lambda$ of the exponential distribution to the denoising timestep (similar to $\bar{\Sigma}_t$), but choose to keep it constant here. As usual in multi-objective optimization, trade-offs between the different loss contributions are expected, depending on the relative weighting (Wang et al., 2021).

## A.6 NUMERICAL STUDIES

For the presented evaluations, we introduce the mean absolute residual error as follows

$$\mathcal{R}_{\mathrm{MAE}}(\boldsymbol{x}_0) = \frac{1}{M+N} \left( \sum_{i=1}^{M} |\mathcal{F}_i[\boldsymbol{x}_0]| + \sum_{j=1}^{N} |\mathcal{B}_j[\boldsymbol{x}_0]| \right), \tag{28}$$

where $i$ and $j$ denote the $i$-th PDE and $j$-th boundary condition constraint, and we consider a total of $M$ and $N$ such constraints, respectively.

### A.6.1 TOY PROBLEM

**Setup.** In a simple, instructive example, we demonstrate the implications of the proposed physics-informed loss function equation 14 both considering the mean and sample estimation for the residual evaluation. The objective is to learn a distribution $q(\boldsymbol{x}_0)$ that samples points uniformly on the unit circle $\mathcal{S}_1 = \{\boldsymbol{\xi} \in \mathbb{R}^2 : \|\boldsymbol{\xi}\| = 1\}$, so that samples should obey a simple algebraic equality constraint. Thus, generated samples of the model $\boldsymbol{x}_0$ directly correspond to the two spatial coordinates $(\xi_1, \xi_2)$. We here identified $c = 0.1$ and $c = 0.005$ via a simple hyperparameter sweep for the mean and sample estimation, respectively, and thus increased the variance of the implied virtual likelihood.

We choose a simple 3-layer MLP of latent dimension 128 with a 2D vector $(\xi_1, \xi_2)$ as in- and output. Information about the diffusion timestep $t$ is added by transforming $t$ to an embedding which is element-wise multiplied by the output of the linear layer, generally followed by the Softplus activation except for the last layer. We train the model for 400 epochs on 10,000 randomly sampled points of the unit circle, using the Adam optimizer (Kingma & Ba, 2014) with a learning rate of $5 \times 10^{-4}$. We use a batch size of 128, and 100 diffusion timesteps with a cosine scheduler (Dhariwal & Nichol, 2021). The training time of each considered model variant takes around 12 minutes on an Nvidia Quadro RTX 6000 GPU (equipped with 24GB GDDR6 memory).

**Results.** To understand the impact of the physics-informed loss, we train the proposed PIDM either via mean or sample estimation in four different scenarios: (i) via the classical setup of the variational bound on the data likelihood equation 4, (ii) via the proposed PIDM loss equation 14, (iii) by only considering the residual loss (and neglecting the variational bound), and (iv) by again considering the PIDM loss equation 14 but with data sampled from an uninformative prior $\boldsymbol{x}_0 \sim \mathcal{N}(\boldsymbol{0}, \boldsymbol{I})$. We present the average residual error of 100 samples generated over 400 training epochs and plot 100 generated samples after training in Figure 5 and Figure 6 for the mean and sample estimation, respectively. More specifically, we provide the mean absolute residual error evaluated as $\mathcal{R}_{\mathrm{MAE}}(\boldsymbol{x}_0) = |\|\boldsymbol{\xi}\|^2 - 1|$, which is averaged over all samples.

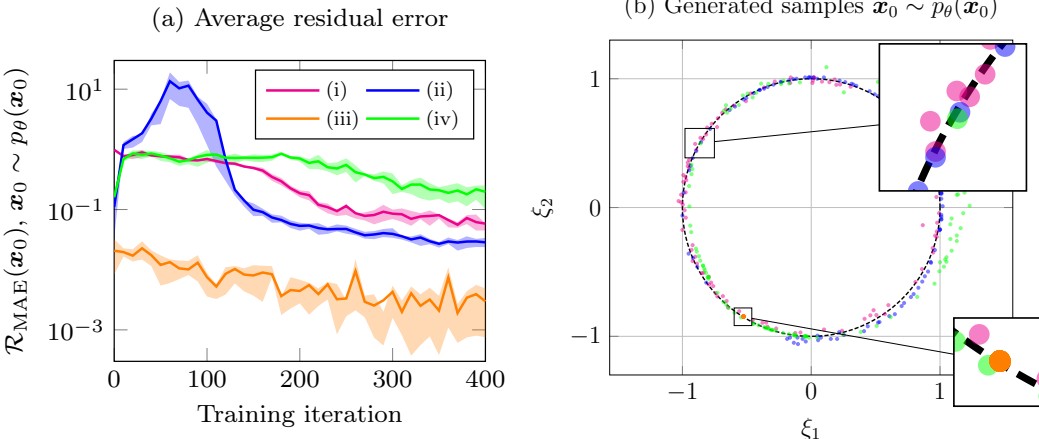

Figure 5: **(Mean estimation)** Evaluation of the average residual error of 100 generated samples during training (a, averaged over 10 training runs with 25/75%-quantiles using different seeds) and 100 generated samples after training (b, from representative models) in four different settings. We consider a diffusion model trained with the standard (data-driven) objective (i), our proposed PIDM (ii), a model trained solely on the residual loss term (iii), and again our proposed PIDM but with data sampled from an uninformative Gaussian prior (iv). The residual during training is evaluated via mean estimation, i.e., $\boldsymbol{x}_0^* = \mathbb{E}[\boldsymbol{x}_0|\boldsymbol{x}_t]$. In (b), we indicate the unit circle to which all samples should be constrained. Colors in (b) match those in (a).

We observe that the studies for the mean and sample estimation present a similar picture, and we may summarize the findings independently of the considered estimation mechanism as follows. As indicated in Figure 5a and 6a, we observe that the proposed physics-informed loss (ii) indeed outperforms the standard setup (i) in terms of the residual error after approximately 100-120 training epochs. This is also visually confirmed in Figure 5b, where the samples from the "constraint-informed"

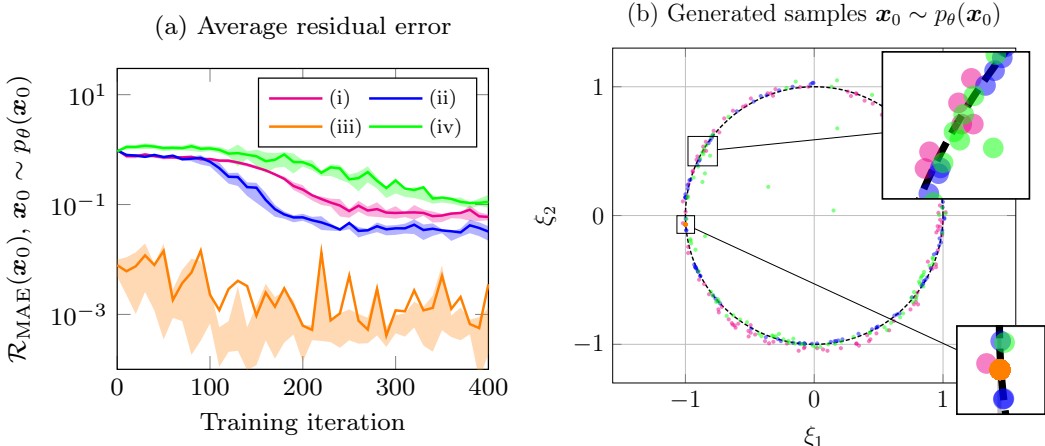

Figure 6: **(Sample estimation)** The setting is identical to the one provided in Figure 5, but we here estimate $\boldsymbol{x}_0^*$ via DDIM (Song et al., 2021a).

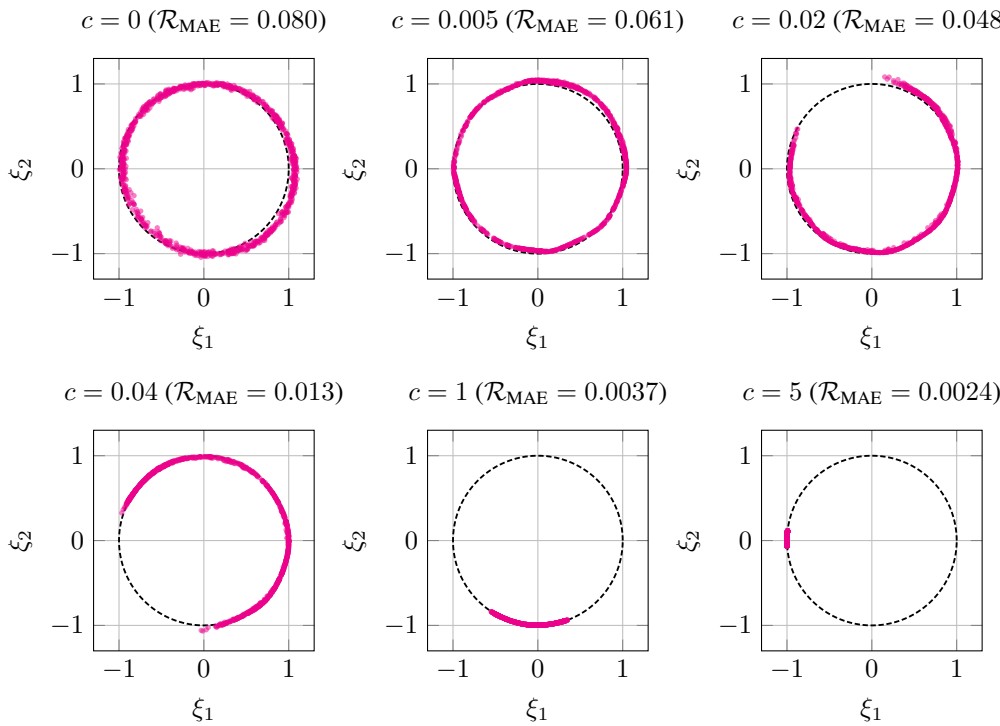

Figure 7: Evaluation of 400 generated samples after training our proposed PIDM with sample estimation in the same setting as Figure 6 but with varying scale factors $c$. We also indicate the mean absolute residual error $\mathcal{R}_{\mathrm{MAE}}$, averaged over all samples and the unit circle to which all samples should be constrained.

model align more accurately on the unit circle (see also the top inset). When training the model solely on the constraints (iii), the model efficiently reduces the residual. However, it converges to a single point randomly located somewhere on $\mathcal{S}_1$, as shown in the bottom inset. This is indeed expected: the model finds no penalty for collapsing to any points on $\mathcal{S}_1$, and similar results are generally observable if the residual penalty vastly exceeds the data loss term. Interestingly, the model can also approximate the target distribution even in the absence of training data using only an uninformative prior (iv), though here at a reduced accuracy. We do not explore this idea further, but

this suggests the use of an uninformative prior as a regularization that prompts the model to explore a wider range of solutions within the constraint space. Although the simple setting of this study prevents straightforward analogies to more complex setups, it confirms the validity of our proposed framework and its ability to enforce constraints.

**Study on the influence of relative weighting.** To systematically examine the effect of the scale factor $c$ that dictates the importance the model assigns to the residual loss, we provide results for six different values in Figure 7. The experimental setup remains consistent with the previous configuration for the sample estimation, except for the variation in $c$. As $c \to 0$, we recover the standard (data-driven) diffusion model. For increasing $c$, the residual error is increasingly reduced, but the model eventually generates samples increasingly concentrated at a point randomly located on the constraint manifold.

As $c \gg 1$, the distribution fully collapses as the model ignores the data loss, and the behavior resembles that of a model trained exclusively on the residual loss (similar to scenario (iii) in Figure 6). For moderate values, such as $c = 0.005$, the distribution achieves an optimal balance: maintaining full diversity while effectively minimizing the residual error.

### A.6.2 DETAILS OF THE DARCY FLOW STUDY

**Background.** The Darcy flow equations describe the steady-state solution of fluid flow through a porous medium. Given a permeability field $K(\boldsymbol{\xi})$, the pressure distribution $p(\boldsymbol{\xi})$ and velocity field $\boldsymbol{u}(\boldsymbol{\xi})$ are governed by

$$
\begin{aligned}
\boldsymbol{u}(\boldsymbol{\xi}) &= -K(\boldsymbol{\xi})\nabla p(\boldsymbol{\xi}), \quad \boldsymbol{\xi} \in \Omega \\
\nabla \cdot \boldsymbol{u}(\boldsymbol{\xi}) &= f(\boldsymbol{\xi}), \quad \boldsymbol{\xi} \in \Omega \\
\boldsymbol{u}(\boldsymbol{\xi}) \cdot \hat{\boldsymbol{n}}(\boldsymbol{\xi}) &= 0, \quad \boldsymbol{\xi} \in \partial\Omega \\
\int_{\Omega} p(\boldsymbol{\xi})\, \mathrm{d}\boldsymbol{\xi} &= 0,
\end{aligned}
\tag{29}
$$

where $\hat{\boldsymbol{n}}$ denotes the outward unit vector normal to the boundary. Similar to contemporary work (Zhu & Zabaras, 2018; Jacobsen et al., 2024), we consider a 2D square domain and set the source function to

$$
f(\boldsymbol{\xi}) = \begin{cases} r, & \text{if } \left|\xi_i - \frac{1}{2}w\right| \le \frac{1}{2}w, \text{ for } i = 1, 2 \\ -r, & \text{if } \left|\xi_i - 1 + \frac{1}{2}w\right| \le \frac{1}{2}w, \text{ for } i = 1, 2 \\ 0, & \text{otherwise}, \end{cases}
\tag{30}
$$

with $r = 10$ and $w = 0.125$. We sample $K(\boldsymbol{\xi})$ from a Gaussian random field (GRF), i.e.,

$$
K(\boldsymbol{\xi}) = \exp(G(\boldsymbol{\xi})), \quad G(\cdot) \sim \mathcal{N}(0, k(\cdot, \cdot))
\tag{31}
$$

with covariance

$$
k\left(\boldsymbol{\xi}, \boldsymbol{\xi}'\right) = \exp\left(-\left\|\boldsymbol{\xi} - \boldsymbol{\xi}'\right\|_2 / l\right), \quad \text{with } l = 0.1.
\tag{32}
$$

Instead of directly considering the GRF, we reduce its dimensionality by considering its Karhunen-Loève expansion up to $s = 64$ terms,

$$
G(\boldsymbol{\xi}) = \sum_{i=1}^{s} \sqrt{\lambda_i} z_i \phi_i(\boldsymbol{\xi}),
\tag{33}
$$

with $\lambda_i$ and $\phi_i(\boldsymbol{\xi})$ as, respectively, the eigenvalues and eigenfunctions of equation 32 sorted by decreasing $\lambda_i$, and $z_i \sim \mathcal{N}(0, 1)$. Discretization of equation 29 is achieved via second-order central finite differences, and we refer to Jacobsen et al. (2024) for details. To incorporate the boundary conditions and the integral constraint, we extend the linear system of type $\boldsymbol{Ap} = \boldsymbol{f}$ by additionally imposing the above constraints, so that $\boldsymbol{A} \in \mathbb{R}^{(n^2+4n+1) \times n^2}, \boldsymbol{p} \in \mathbb{R}^{n^2}$, and $\boldsymbol{f} \in \mathbb{R}^{n^2+4n+1}$ (where $n = 64$). We solve for the over-determined pressure field using the `scipy.linalg.lstsq` (Virtanen et al., 2020) solver with default settings.

**Model architecture and residual computation.** The considered U-Net (Ronneberger et al., 2015) architecture is based on Wang (2022), given its demonstrated success in learning the denoising process (Dhariwal & Nichol, 2021). Our configuration uses an image input resolution of $64 \times 64$

pixels, aligning with the grid resolution of the linear system under consideration. Importantly, we require the residual evaluation of the predicted images $\mathcal{R}(x_0^*)$, as required in equation 14, to be consistent with the data, as otherwise the optimal data likelihood is in partial conflict with the (virtual) residual likelihood. This is ensured by using the same finite difference stencils as in the dataset creation, essentially reassembling $f$, except that we remove the integral constraint, since it can be trivially fulfilled by shifting the predicted pressure field (Jacobsen et al., 2024). Finite difference stencils are implemented via `torch.nn.Conv2D` (Paszke et al., 2019) with a custom kernel, which we can precompute for stencils up to arbitrary order via `findiff` (Baer, 2018).

We here restrict ourselves to an unconditional model, though extensions to conditional generation (Ho & Salimans, 2022) are straightforward. The U-Net has two in- and output channels and is trained to predict the clean signal based on a noisy input, as stated in Section 2.1. Thus, the model is trained to generate pairs $(K, x)$ where $K$ is sampled similar to equation 31, and $p$ is the corresponding (unique) pressure field that satisfies the Darcy flow equations equation 29. The residual is then assembled by considering $\mathcal{F}[x_0] := \nabla \cdot (K\nabla p) + f = 0$. Further details of the model architecture and training hyperparameters are given in Appendix A.7 and the code.

**Comparison with other frameworks.** Note that our architecture of the physics-guided diffusion model (ii) is not an exact replication of the one given by Shu et al. (2023) due to the different setting, but the proposed conditioning mechanism is closely mimicked, as detailed in the code. Jacobsen et al. (2024) presented a strategy to iteratively "correct" the latent variables $x_t$ and samples $x_0$ during inference by applying gradient descent based on the PDE residuals (iii). We first followed the optimal setting proposed in Jacobsen et al. (2024) and applied gradient-based descent $\nabla_{x_t}\|\mathcal{R}(x_t)\|_2^2$ with $\epsilon = 2 \times 10^{-4}/\max \nabla_{x_t}\mathcal{R}(x_t)$ for the last $N$ steps of the sampling iterations and $M$ additional iterations. We set $M = 25$ and $N = 50$ according to the best-reported results (equal to starting corrections halfway through the sampling). However, we encountered stability issues for the above value of $\epsilon$, likely due to differences in the sampling scheme and fewer considered timesteps compared to Jacobsen et al. (2024), who adopted a score-based perspective. We therefore conducted a parameter sweep to identify the converging results with the best performance, which we identified at $\epsilon = 1 \times 10^{-6}/\max \nabla_{x_t}\mathcal{R}(x_t)$. While additional sweeps might yield some further incremental improvement, we observed that different values of $\epsilon$ yielded similar results, as long as the updates converged.

### A.6.3 DETAILS OF THE TOPOLOGY OPTIMIZATION STUDY

**Background.** Topology optimization aims to identify a structure with optimal mechanical properties, typically optimal mechanical stiffness. This can be formalized as the minimization of the mechanical compliance C under a set of equality constraints (given by mechanical equilibrium and boundary conditions) and inequality constraints (typically a volume constraint). Under the assumptions of linear elasticity, the problem reads

$$\min_{\rho(\boldsymbol{\xi})} \underbrace{\int_\Omega \frac{1}{2}\boldsymbol{\sigma}(\boldsymbol{\xi}) : \varepsilon(\boldsymbol{\xi})\,\mathrm{d}\Omega}_{C}, \quad \boldsymbol{\xi} \in \Omega$$

$$\text{subject to:} \quad \rho(\boldsymbol{\xi}) \in [0, 1],$$

$$\int_\Omega \rho(\boldsymbol{\xi})\,\mathrm{d}\Omega \leq V_{\max},$$

$$\nabla \cdot \boldsymbol{\sigma}(\boldsymbol{\xi}) + \boldsymbol{f}(\boldsymbol{\xi}) = \mathbf{0}. \tag{34}$$

The last equation implies quasistatic mechanical equilibrium. Here, $\varepsilon$ and $\boldsymbol{\sigma}$ denote the strain and stress tensor fields, respectively, $\boldsymbol{f}$ is a distributed body force and $V_{\max}$ a given (maximum) volume constraint. We consider a linear elastic material, which couples $\varepsilon$ and $\boldsymbol{\sigma}$ via Hooke's law, $\boldsymbol{\sigma} = \boldsymbol{C} : \varepsilon$, where $\boldsymbol{C}$ is the fourth-order stiffness tensor, and the colon denotes double tensor contraction. For simplicity, we may assume an isotropic material, so $\boldsymbol{C}$ is characterized by two material constants (e.g., Young's modulus $E$ and Poisson's ratio $\nu$). The solution field is given in terms of the displacements $\boldsymbol{u}(\boldsymbol{\xi})$, from which the strain tensor follows as $\varepsilon = \frac{1}{2}[\nabla\boldsymbol{u} + (\nabla\boldsymbol{u})^{\mathsf{T}}]$. Dirichlet boundary conditions are applied as $\boldsymbol{u} = \bar{\boldsymbol{u}}$ on the boundary $\partial\Omega_u \subset \Omega$, and traction boundary conditions $\boldsymbol{\sigma} \cdot \hat{\boldsymbol{n}} = \boldsymbol{t}$ on $\partial\Omega_t \subset \Omega$, where $\hat{\boldsymbol{n}}$ denotes the outward unit vector normal to the boundary $\subset \Omega$.

In practice, equation 34 is usually discretized via finite elements. We here consider a regular finite element mesh, based on a $65 \times 65$ grid of nodes with ($64 \times 64$) four-node quadrilateral elements under plane-stress assumptions (with $E = 1$, $\nu = 0.3$). This turns the mechanical equilibrium equation into a linear system of type $\boldsymbol{KU} = \boldsymbol{F}$, where $\boldsymbol{K} \in \mathbb{R}^{2n^2 \times 2n^2}$ is the global stiffness matrix and $\boldsymbol{U}, \boldsymbol{F} \in \mathbb{R}^{2n^2}$ are the global nodal displacement and the external force vectors, respectively, with $n = 65$. Dirichlet boundary conditions are imposed by appropriately modifying $\boldsymbol{K}$ and $\boldsymbol{F}$. Optimized topologies can subsequently be obtained via the Solid Isotropic Material with Penalization (SIMP) method (we refer to Bendsøe & Sigmund (2004) for details). SIMP considers continuous densities in equation 34 to allow for gradient-based optimization, defining $\boldsymbol{C}(\boldsymbol{\xi}) = \rho(\boldsymbol{\xi})^p \boldsymbol{C}^0$, where $p > 1$ promotes binary entries (corresponding to material placement, $\rho = 1$, or void, $\rho = 0$) and $\boldsymbol{C}^0$ denotes the material properties of the given isotropic base material. As SIMP often requires many costly finite element analyses to iteratively refine the solution and may get stuck in local minima, deep learning frameworks including generative adversarial networks (Nie et al., 2021) and diffusion models (Mazé & Ahmed, 2023; Giannone et al., 2023) have been explored as alternatives to mitigate some of these challenges.

**Model architecture and residual computation.**  We consider the same U-Net architecture as in Section 4.1 except for the following differences. Similar to Nie et al. (2021), we also provide the von Mises stress and strain energy fields of the unoptimized domain, as well as the boundary conditions (including the loads) and volume fraction to allow for conditioning in addition to the noisy signal of the solution fields (consisting of the two displacement and density fields) to the model. The model aims to reconstruct the clean signal of the two displacement fields $\boldsymbol{u}_{1,2}$ and the optimal density $\boldsymbol{\rho}$. Besides, we increase the latent dimension of the U-Net to 128 to have a similar number of overall parameters compared to previous work (Mazé & Ahmed, 2023; Giannone et al., 2023). Lastly, we apply a sigmoid activation after the density field channel output to ensure $\rho \in [0, 1]$. The predicted $\rho(\boldsymbol{\xi})$ then enters the governing equations via $\boldsymbol{C}(\boldsymbol{\xi}) = \rho(\boldsymbol{\xi}) \boldsymbol{C}^0$.

For the residual and compliance computation, using finite differences as in Section 4.1 to assemble equation 34 introduces inconsistencies with the FEM solution (i.e., training data with non-zero residuals), likely due to the sharp transitions in the density field and point-wise introduction of loads. This is problematic, as the minimization of the residual then does not fully correspond to samples from the data distribution, leading to an optimization conflict. We hence treat the pixels as direct representations of the FE grid and apply the stiffness matrix $\boldsymbol{K}$ to evaluate the residual. As we consider a U-Net with in- and output pixel dimensions of $64 \times 64$, we apply bilinear interpolation to down- or upscale all nodal quantities to a $65 \times 65$ grid where necessary. Note that we can obtain the global stiffness matrix efficiently by precomputing the local stiffness matrix (up to a factor depending on the corresponding density $\rho(\boldsymbol{\xi})$ at the element level) given, e.g., by `SolidsPy` (Gómez & Guarín-Zapata, 2018) and vectorizing the global assembly. Further details of the model architecture and training hyperparameters are given in Appendix A.7 and the code.

**Evaluation.**  We present metrics consistent with Mazé & Ahmed (2023); Giannone et al. (2023), namely, the compliance error is introduced as $\text{CE} = (\text{C}(\boldsymbol{x}_0) - \text{C}(\boldsymbol{x}_{0,\text{SIMP}}))/\text{C}(\boldsymbol{x}_{0,\text{SIMP}})$ and the volume fraction error as $\text{VFE} = |\bar{\rho}(\boldsymbol{x}_0) - \rho_{\text{target}}|/\rho_{\text{target}}$, where $\bar{\rho}$ is the (binarized) density averaged over the domain. All models use 100 denoising steps to generate a sample. We note that the models provided by Mazé & Ahmed (2023); Giannone et al. (2023) are trained for only 200k iterations but with a 16 times larger batch size (64). Besides, these frameworks require additional overhead due to the generation of auxiliary data and training of potential surrogate models (Mazé & Ahmed, 2023). For the comparison with PG-Diffusion (Shu et al., 2023) and CoCoGen (Jacobsen et al., 2024) we included both the mechanical equilibrium equations and the volume constraint in the residual. For CoCoGen, computing the full Jacobian $\nabla_{\boldsymbol{x}_t} \boldsymbol{\mathcal{R}}(\boldsymbol{x}_t)$ (here, $\boldsymbol{\mathcal{R}}(\boldsymbol{x}_t)$ is vector-valued), which the authors use to scale the residual exceeds the VRAM of our GPU (even with a batch size of 1 and efficient implementation via `torch.func.jacfwd`). We thus considered a constant scaling factor, conducted an extensive hyperparameter study, and report the best results (in terms of the median residual on the in-distribution test set).

### A.7 U-NET ARCHITECTURE AND TRAINING DETAILS

As described in Section 4.1 and 4.2 we consider a U-Net-based architecture (Ronneberger et al., 2015). The main model and training hyperparameters are summarized in Tables 2 and 3, respectively. The model is implemented and trained using PyTorch (Paszke et al., 2019). Further details can be found in the code.

For the Darcy flow study, training for 300k iterations took approximately 13 hours for the standard diffusion setup, 16 hours for the PIDM with mean estimation, and 22 hours for the PIDM with sample estimation. For the topology optimization study, training took approximately 48 hours for the standard diffusion setup and 54 hours for the PIDM (with sample estimation). All models were trained on a single Nvidia Quadro RTX Quadro RTX 6000 GPU equipped with 24GB GDDR6 memory.

Table 2: Denoising diffusion architecture hyperparameters.

| Hyperparameter | Value |
|---|---|
| In-, output channels *(Darcy flow)* | 2, 2 |
| In-, output channels *(Topology optimization)* | 10, 3 |
| ResNet blocks per down- and upsampling pass | 2 |
| ResNet block normalization | Group Normalization (Wu & He, 2018) |
| ResNet block activation function | SiLU (Elfwing et al., 2018) |
| Attention block normalization | Layer Normalization (Ba et al., 2016) |
| Feature map resolutions (downsampling pass) | $64 \times 64 \to 32 \times 32 \to 16 \times 16 \to 8 \times 8$ |
| Latent dimensions (in feature maps, *Darcy flow*) | $32 \to 64 \to 128 \to 256$ |
| Latent dimensions (in feature maps, *Top. opt.*) | $128 \to 256 \to 512 \to 1024$ |
| Attention (Vaswani et al., 2017; Katharopoulos et al., 2020) head dimension | 32 |
| Number of attention heads | 8 |

Table 3: Denoising diffusion process and training hyperparameters.

| Hyperparameter | Value |
|---|---|
| Number of diffusion timesteps | 100 |
| $\beta_t$-scheduler | Cosine schedule (Dhariwal & Nichol, 2021) |
| Batch size (*Darcy flow*), mean estimation | 64 |
| Batch size (*Darcy flow*), sample estimation | 16 |
| Batch size (*Topology optimization*, sample estimation) | 4 |
| Iterations (*Darcy flow*) | 300k |
| Iterations (*Topology optimization*) | 600k |
| Learning rate | $10^{-4}$ |
| Optimization algorithm | Adam (Kingma & Ba, 2014) |
| EMA start (iteration) | 1,000 |
| Exponential Moving Average (EMA) decay | 0.99 |
| Dropout | none |

## A.8 ADDITIONAL SAMPLES

### A.8.1 DARCY FLOW

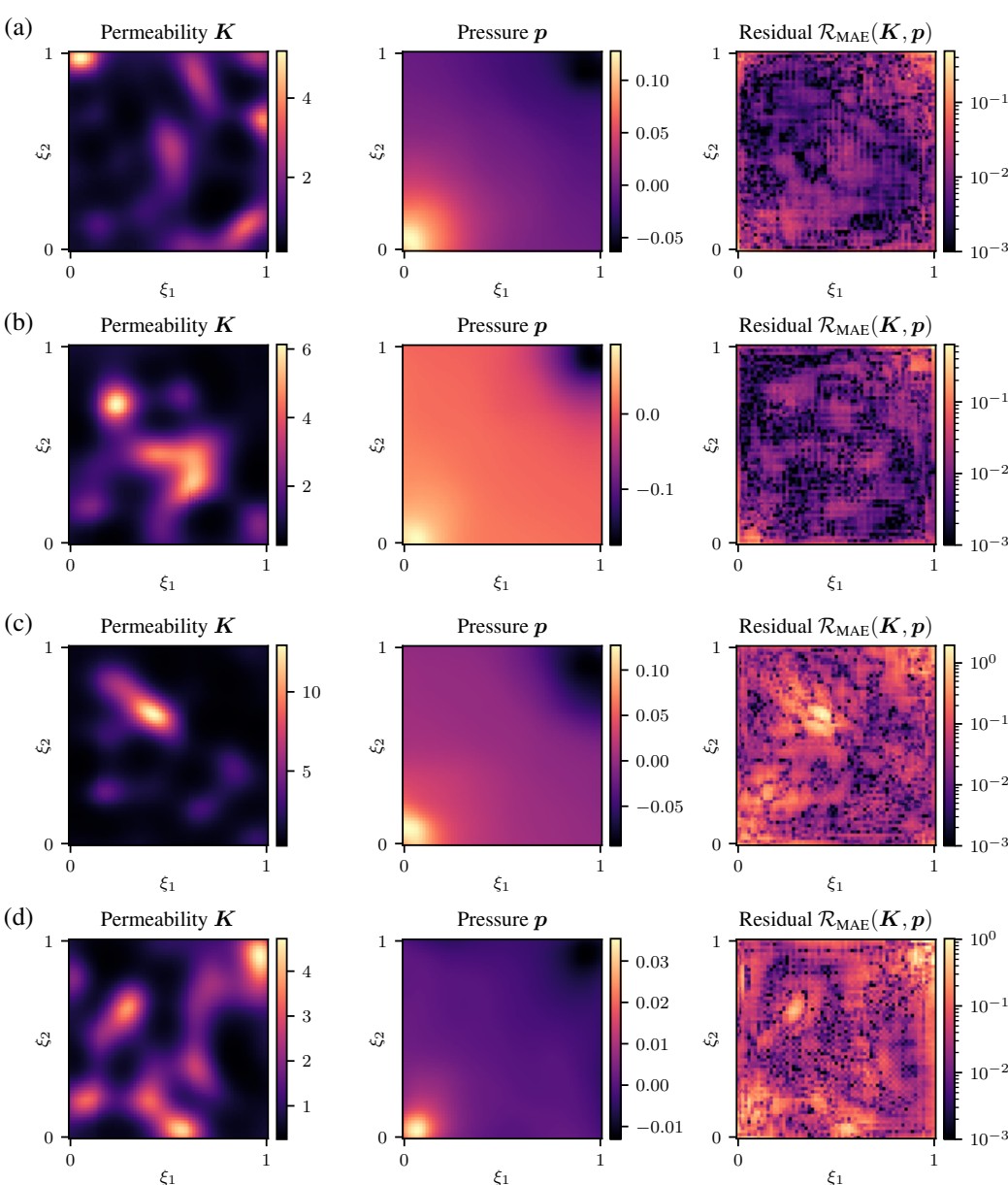

Figure 8: Additional samples of permeability and pressure fields as well as the corresponding residual error from the proposed PIDM trained on the Darcy flow dataset. Sample (a) and (b) are sampled from a PIDM with mean estimation, while (c) and (d) are sampled from a PIDM with sample (DDIM) estimation.

### A.8.2 TOPOLOGY OPTIMIZATION

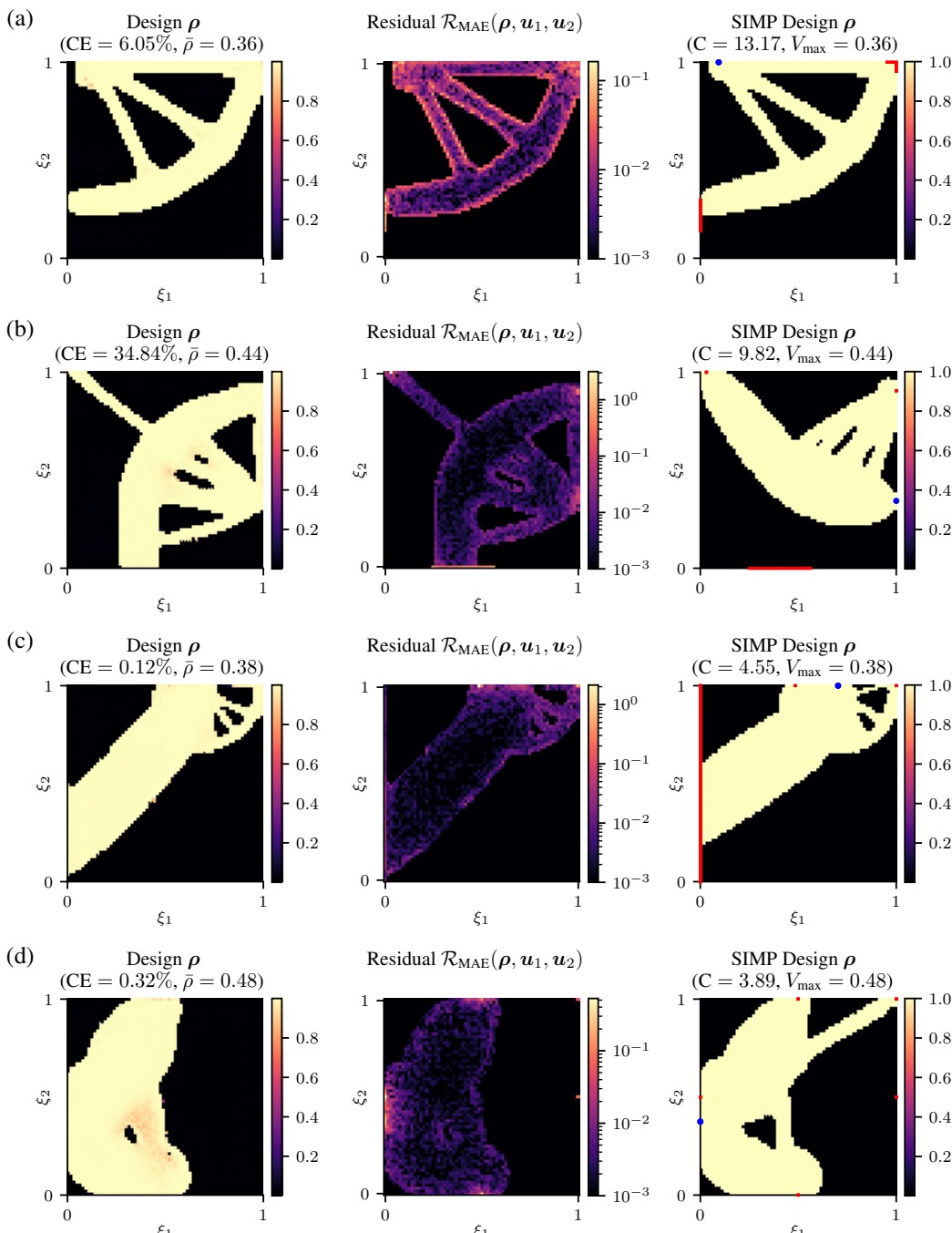

Figure 9: Additional generated designs, including the compliance error CE and volume $\bar{\rho}$, and the residual error (based on the displacement fields, not shown) from the PIDM trained on the SIMP dataset and the corresponding SIMP design, including the compliance C and volume $V_{\max}$. All samples are conditioned on the out-of-distribution test set. In the SIMP design, we indicate the applied load by a blue dot, and the Dirichlet boundary conditions in red.

