# OpenReview forum: "Physics-Informed Diffusion Models"
_ICLR.cc/2025/Conference — ICLR 2025 Poster_

### Official Review · Reviewer_oHnR · 2024-10-17

**Soundness:** 2
**Presentation:** 3
**Contribution:** 2
**Rating:** 5
**Confidence:** 3

**Summary:**

This paper employs Diffusion Models to satisfy constraints, with applications to Partial Differential Equations (PDEs) and Topology Optimization. The diffusion model is trained using a combination of a data-free loss term and a data-driven diffusion loss.

**Strengths:**

The setting of combining a data-based diffusion loss with a data-free diffusion loss in general is an interesting question.

**Weaknesses:**

1. The core contribution of this paper is the use of Diffusion Models to sample from an unnormalized target distribution. However, there is extensive prior work on this topic (see [1], [2], [3], [4]), which the authors have not cited. The paper's approach includes an additional Physics-Informed Neural Network (PINN) regularization loss, which can be considered a special case of Eq. 6 in Sanokowski et al. [1], where the temperature is set to $0$. The authors of [1] emphasize that temperature regularization, combined with annealing, is crucial for avoiding local minima (see Fig. 2, middle panel, in their paper). Thus, the current approach may perform poorly in the absence of a supervised loss term.

   Additionally, the "Noise Distribution" described in Eq. 12 of the paper is essentially the "Annealed Noise Distribution" introduced in Sec. 3.2 of [1]. The authors should also cite [5], which discusses training neural networks with PINN losses in a generative manner without using any data.

   The authors could address these concerns by properly citing these works in the related literature. Furthermore, the proposed method might benefit from adopting the loss function used in [1] and incorporating annealing during training. It would be interesting to study the combination of a data-driven loss with a data-free annealing-based loss.

2. Experimental Evaluation:
The experimental evaluation is limited and should be expanded to cover a broader range of settings.

### References
[1] Zhang, Qinsheng, and Yongxin Chen. "Path Integral Sampler: A Stochastic Control Approach For Sampling." International Conference on Learning Representations.

[2] Berner, Julius, Lorenz Richter, and Karen Ullrich. "An Optimal Control Perspective on Diffusion-Based Generative Modeling." Transactions on Machine Learning Research.

[3] Vargas, Francisco, Will Sussman Grathwohl, and Arnaud Doucet. "Denoising Diffusion Samplers." Eleventh International Conference on Learning Representations.

[4] Sanokowski, Sebastian, Sepp Hochreiter, and Sebastian Lehner. "A Diffusion Model Framework for Unsupervised Neural Combinatorial Optimization." Forty-First International Conference on Machine Learning.

**Questions:**

1. **Clarification Needed:**
   The paper states:
   > "Since the remaining derivations hold true for both estimation techniques, we introduce \(x_0^*\) to denote either the mean or the sample estimate on which the residual is evaluated."

   The rationale for introducing \(x_0^*\) is unclear. What advantage does this provide? Why not directly sample \(X_0\) from the model? How is \(x_0^*\) computed?

Can you make an ablation on this choice?

2. How does your method perform when the data-based loss term is removed?

---

> ### Author Response · Authors · 2024-11-24
> **Clarifying the setting of PIDMs**
>
> We thank the reviewer for their critical evaluation of our method and for providing a list of references on leveraging diffusion models to address sampling problems.
>
> **Clarification of core contribution**
>
> We would strongly like to clarify that mere sampling from an unnormalized target distribution is *not the setting in which we operate*. Instead, we address the common setting that involves leveraging the expressivity of diffusion models on available datasets in the natural and engineering sciences. For only a selection of recent works that operate in this exact setting, we have provided the references *Xie et al., 2021; Buehler, 2022; Bastek & Kochmann, 2023; Vlassis & Sun, 2023; Sardar et al., 2023; Li et al., 2023; Lienen et al., 2023* in the manuscript. Unlike the references provided by the reviewer, our setting assumes the availability of data, which fundamentally distinguishes it from the sampling-focused approaches highlighted by the reviewer. (Moreover, Sanokowski et al. operate in a combinatorial space, whereas we consider continuous solution fields, raising questions about the applicability of their results to our setting.)
>
> Concretely, our model is not designed to learn all possible samples that fulfill a constraint but rather to augment data-driven diffusion models by grounding them in the known physical principles that govern the available data. Adopting a purely residual-based approach, closer to the references provided, would require the diffusion model to primarily *function as a numerical solver and thus compete with decades of research into highly specialized solvers for the corresponding PDEs*. While this setup is possible, we do not believe this to be a realistic or practical use case, and note that this might also explain why none of the provided references address PDE-based systems.
>
> Additionally, we emphasize that all theoretical insights are derived under the assumption of this specific (data-driven) setting. In Appendix A.1, we demonstrate how our augmented loss continues to recover the data distribution, and in Appendix A.3, we formulate a simplified loss based on the assumption that samples from the data distribution are available. This approach aligns with prior work aimed at grounding diffusion models in physical principles in scientific applications, such as [1-4], to which we have extensively compared our framework. We hope the reviewer acknowledges this critical distinction and reconsiders their negative evaluation.
>
> **Question 1:** As highlighted in Section 3.3, a primary challenge in diffusion models is that directly sampling $\boldsymbol{x}_0$ is computationally infeasible. A significant contribution of our work is the development of two frameworks—both outperforming the state of the art in residual error reduction—that circumvent this complexity. These frameworks either use the (straightforwardly available) mean estimate or an accelerated sample estimate that requires only one additional forward pass. We provide theoretical arguments demonstrating the consistency of the latter approach, which is experimentally validated in Figure 2c.
>
> Since the subsequent steps we introduce remain independent of using either the mean or the sample estimate, we simply introduced $\boldsymbol{x}_0^*$ as a notation that may relate to both estimates. We acknowledge the remark from the reviewer that this may be unclear and have clarified this in the revised manuscript.
>
> **Question 2:** We have provided studies on the effect of removing the data-based loss term in Appendix A.6.1 (Figure 5, 6; scenario (iii)). In this case, the distribution simply collapses to a random point on the constraint manifold and resembles the training of a classical PINN. We observed a similar behavior in the fluid flow case study but did not include it, as, again, this is not the scenario we envision with PIDMs.
>
> [1] Dule Shu, Zijie Li, and Amir Barati Farimani. A physics-informed diffusion model for high-fidelity
> flow field reconstruction. Journal of Computational Physics, 478:111972, 2023.
>
> [2] Christian Jacobsen, Yilin Zhuang, and Karthik Duraisamy. CoCoGen: Physically-Consistent and
> Conditioned Score-based Generative Models for Forward and Inverse Problems.
>
> [3] François Mazé and Faez Ahmed. Diffusion Models Beat GANs on Topology Optimization. Proceedings of the 37th AAAI Conference on Artificial Intelligence, AAAI 2023, 37:9108–9116, 2023.
>
> [4] Giorgio Giannone, Akash Srivastava, Ole Winther, and Faez Ahmed. Aligning Optimization Trajectories with Diffusion Models for Constrained Design Generation. Advances in Neural Information Processing Systems, 2023.

---

> > ### Comment · Reviewer_oHnR · 2024-11-25
> > **Answer to Rebuttal**
> >
> > I must respectfully disagree with your main points:
> >
> > Regarding your first clarification: **We would strongly like to clarify that mere sampling from an unnormalized target distribution is not the setting in which we operate.**
> >
> > This statement appears inconsistent with your methodology. In line 147, you explicitly describe part of your problem as sampling residuals from the target distribution defined in Equation 7. When implemented without data, this process is precisely equivalent to sampling from an unnormalized target distribution. While your approach does incorporate an additional data-based loss term (as acknowledged in my initial review), this addition does not fundamentally alter the underlying sampling framework.
> >
> > Concerning your second point: **Moreover, Sanokowski et al. operate in a combinatorial space, whereas we consider continuous solution fields, raising questions about the applicability of their results to our setting**
> >
> > This distinction is less substantial than suggested. The framework in Sanokowski et al. can be naturally extended to continuous field settings. Your Equation 12 effectively implements Sanokowski's "Annealed Noise Distribution" in continuous space, but without temperature regulation. Moreover, your experiments include Topology Optimization Problems, which are inherently discrete problems similar to those in Sanokowski et al. (2024). While these can be solved through continuous relaxation, this actually demonstrates the connection to Sanokowski's framework rather than distinguishing it.
> >
> > Regarding **Question 2: We have provided studies on the effect of removing the data-based loss term in Appendix A.6.1 (Figure 5, 6; scenario (iii)). In this case, the distribution simply collapses to a random point on the constraint manifold and resembles the training of a classical PINN. We observed a similar behavior in the fluid flow case study but did not include it, as, again, this is not the scenario we envision with PIDMs.**
> >
> > The behavior you observe directly aligns with my expectations for systems lacking temperature regulation, as described in Sanokowski et al. This is why I would encourage the authors to use an entropy regularization term to make the method more robust to more complicated settings or to settings where less data is available.
> >
> > In conclusion, given the limited experimental validation and substantial methodological overlap with existing approaches which are also not cited in this paper, I do not believe this work meets the criteria for top-tier conference publication.

---

> ### Author Response · Authors · 2024-11-27
> **Follow-up clarification on answer to rebuttal**
>
> We would first like to thank the reviewer for taking the time to go through our response, providing timely feedback, and engaging in this interesting discussion. We appreciate the opportunity to clarify the contributions and novelty of our work.
>
> We agree with the reviewer that Equations 7–9, when viewed on their own, can be understood as sampling from an unnormalized distribution constructed by evaluating $q_{\boldsymbol{\mathcal{R}}}$ over $\boldsymbol{x}_0$. This approach aligns with the setting introduced by Rixner & Kourotsekalis (2021), whom we have cited in the relevant parts of our paper. As we acknowledge by referencing them, this is not our key contribution. Again, we emphasize that without considering data, we compete against highly optimized numerical solvers designed for the specific choices of PDE.
>
> Our key contribution follows afterward, where we augment the common data-driven setup of diffusion models to improve adherence to the set of PDEs under consideration. Unlike the sampling problems as given by the reviewer, we train the model *not* by sampling from a simple prior; instead, we query our model on noisy latents obtained from the data points, changing training dynamics as the model operates on latents that are aligned with the true distribution. More importantly, a key contribution of our framework lies in identifying **simple, but effective surrogates to obtain the clean signal at any point in the diffusion trajectory**, for which we have provided both the so-called **mean** and **sample estimation**. Both methods successfully outperform prior SOTA in terms of residual fulfilment. By contrast, all references of the reviewer require sequentially sampling over the whole diffusion trajectory to obtain a sample based on which the energy or unnormalized density is evaluated. This increases training time by orders of magnitude and also has significantly larger memory requirements than the standard (data-driven) score matching techniques (to which we count our approach).
>
> To our knowledge - and we thank the reviewer for pointing out this interesting connection we were not aware of - the only partial exception is indeed the 'annealed noise distribution' by Sanokowski et al. Nevertheless, **we would like to clarify that this setup is fundamentally different to our equation 12**. As mentioned in the prior paragraph, a key insight is that **we evaluate our residual on the mean or sample estimate** of the clean signal, while Sanokowski et al. evaluate it on the **latent** (i.e., $X_t$ ($\boldsymbol{x}_t$ in our notation), see bottom right of page 4 in Sanokowski et al.). In fact, we have pointed out in Appendix A.1 that evaluating the residual on such latents is not consistent: Adding noise to the solution field drastically increases the residual error due to its high sensitivity to higher-order derivatives of the solution fields, rendering this loss ineffective. Early investigations confirmed this issue (which we are happy to reproduce if it is of interest to the reviewer). Instead, we think it could be of interest to apply our estimation techniques for $\boldsymbol{x}_0$ to their setting.
>
> Regarding topology optimization, while the material distribution is indeed a continuous relaxation of a binary design space, we highlight that the physical fields (i.e., displacements), which dictate the performance of the design and are part of the model predictions, are not discrete. Also, we clarify that we have in fact considered a form of entropy regularization by considering Gaussian noise as data, as briefly investigated in Figure 5, 6, case (iv), and observe that this encourages the model to obtain diverse results that still respect the constraint.
>
> Lastly, we point out that we have presented two studies in relevant fluid and structural mechanical settings (e.g., [1] solely focuses on the latter setting and was published in a similar top-tier conference), besides a toy study (which is similar to the experiments shown in some of the references the reviewer has provided). While we agree that additional studies would further showcase the applicability of PIDMs, we disagree with the notion of a 'limited experimental validation'.
>
> Once again, we would like to thank the reviewer for pointing out the interesting connections. We have included the suggested references in the appropriate section (3.1) in our revised manuscript. At the same time, we would greatly appreciate it if the reviewer could reevaluate their assessment of the 'substantial methodological overlap' given the clear theoretical and technical distinctions listed above.
>
> [1] Giorgio Giannone, Akash Srivastava, Ole Winther, and Faez Ahmed. Aligning Optimization Trajectories with Diffusion Models for Constrained Design Generation. Advances in Neural Information Processing Systems, 2023.

---

> ### Comment · Reviewer_oHnR · 2024-11-29
> **Raised score**
>
> I appreciate the authors' clarifications regarding their methodology. Upon careful reconsideration, I acknowledge that I previously overlooked their approach of rewriting the expectation over model samples to an expectation over data samples by assuming uniform importance weights. In light of this understanding, I am revising my score to 5.
>
> Since the authors are exclusively working with expectations over data samples, I believe their method is working in a similar setting as the "Guidance methods" of Diffusion models, such as the work by Yu et al. ("Freedom: Training-free energy-guided conditional diffusion model"). To strengthen the paper, I recommend conducting a comparison with existing guidance-based methods.
>
> Additionally, I suggest the authors include a reference to Sanokowski et al., due to the notable similarity between Equation 12 in this paper and the annealed noise distribution presented in their work.
>
> Yu, Jiwen, et al. "Freedom: Training-free energy-guided conditional diffusion model." Proceedings of the IEEE/CVF International Conference on Computer Vision. 2023.

---

> > ### Author Response · Authors · 2024-11-30
> > **Follow-up on Official Comment**
> >
> > We sincerely appreciate the reviewer’s reconsideration of their rating of our framework.
> >
> > Since the deadline for adjusting the manuscript has passed, we unfortunately cannot update the reference to the best of our knowledge. If given the opportunity, we are happy to provide it in conjunction with Equation 12. To further clarify as far as restrictions on anonymity allow, the reason for not including the reference is that our preprint was publicly accessible months before Sanokowski et al.
> >
> > The reviewer correctly points out relevant connections to guidance-based methods, such as those introduced by Yu, Jiwen, et al. (2023). While these methods share similar objectives, their underlying methodology is fundamentally distinct from ours: such methods are not applied during training and, therefore, do not address the underlying distribution learned by the model. Instead, they function as elaborate post-processing techniques. As we state in our manuscript, these approaches could be used in conjunction with our method since we do not alter the inference process.
> >
> > More importantly, we **do indeed compare our method to such guidance-based approaches**. Specifically, CoCoGen, as introduced in [1], operates in a setting very similar to that of Yu, Jiwen, et al. (2023). In [1], Equation 21 corresponds to the 'energy diffusion guidance' (Equation 5) in Yu, Jiwen, et al. (2023). In our manuscript, we extensively compare PIDMs to CoCoGen and demonstrate that PIDMs significantly outperform it in the exact same setting proposed in [1] (see Fig. 2 of our manuscript).
> >
> > We highlight that Yu, Jiwen, et al. (2023) indeed propose applying guidance based on evaluating the energy function on the *mean estimate of the clean signal* $\mathbb{E}(\boldsymbol{x}_0\vert\boldsymbol{x}_t)$ (Equation 7), rather than the *latent* $\boldsymbol{x}_t$ (which is the setting of CoCoGen and Sanokowski et al., as discussed in our earlier response). During the early stages of our work, we explored similar ideas and implemented a guidance-based method that corrects the latents not by evaluating the residual of $\boldsymbol{x}_t$ but also $\mathbb{E}(\boldsymbol{x}_0\vert\boldsymbol{x}_t)$. This exact setting can in fact be toggled using the `correction_flag` in `model.yaml` provided in our code.
> >
> > However, our experiments revealed that **guidance corrections based on the mean estimate did not significantly improve residual fulfillment in the Darcy flow setting** (which is why we continued to focus on evaluations consistent with the variant employed in CoCoGen [1]), and providing this correction during training (not inference) is crucial.
> >
> > Beyond these differences, we emphasize that the use of accelerated sampling strategies via DDIM [2] to achieve more consistent sample estimates *during training* (not inference!) is, to the best of our knowledge, a novel contribution.
> >
> > Therefore, while we greatly appreciate the revised score, we remain concerned by the assessment of our contributions as 'poor' and the overall rating, which continues to lean toward rejection despite our thorough responses to the points raised. We thank the reviewer for their extensive time and effort, and we would greatly appreciate further reconsideration.
> >
> > [1] Christian Jacobsen, Yilin Zhuang, and Karthik Duraisamy. CoCoGen: Physically-Consistent and Conditioned Score-based Generative Models for Forward and Inverse Problems, 2023.
> >
> > [2] Jiaming Song, Chenlin Meng, and Stefano Ermon. Denoising Diffusion Implicit Models. International Conference on Learning Representations, 2021.

---

> > > ### Author Response · Authors · 2024-12-02
> > > **Requesting the reviewer for feedback**
> > >
> > > With the discussion period coming to a close, we would greatly appreciate any additional feedback on our rebuttal. We sincerely thank the reviewer for the time they have already dedicated to addressing it and remain available to provide further clarifications or answer any additional questions regarding the paper or our response.

---

> ### Comment · Reviewer_oHnR · 2024-12-02
> **Answer to request for feedback**
>
> I agree with the authors that their initial contribution rating of 1 does not adequately reflect their work. Therefore, I have increased the contribution score to 2.
> However, I will maintain my current overall score of 5 because I consider the scope of experiments and the contribution to be small.
> Finally, I noticed that on lines 29, 379, and 397, question marks (?) have been inserted where proper citations should appear. These sections require appropriate academic references to be added.

---

> ### Author Response · Authors · 2024-12-02
> **Answer to reviewer**
>
> We appreciate the reviewer's reconsideration, though we respectfully disagree on the small scope of the experiments as we have indeed provided 'a comparison with existing guidance-based methods', as requested by the reviewer and evident in the supplementary code, and other frameworks operating in this setting. We thank the reviewer for identifying the lines that were not properly compiled and will ensure these issues are resolved in the final version upon acceptance.

---

### Official Review · Reviewer_VMKn · 2024-11-03

**Soundness:** 3
**Presentation:** 3
**Contribution:** 3
**Rating:** 6
**Confidence:** 4

**Summary:**

The authors proposed a new strategy to generate physical data under the physical constraints by directly combing the training target of diffusion models and the physical PDEs in the loss function. The data used in PDEs are estimated via one-step score-based mean estimation or two-step DDIM sampling. The authors compared their method to other physical-guided machine learning models in the darcy flow generation and topology optimization tasks, which shows their models outperformed others in conforming to the physical constraints but underperformed in other criteria.

**Strengths:**

Compared to previous methods [1][2] that integrates physical constraints by projecting the PDE residuals to the latent space, the authors introduced the PDE residuals as virtual observables and explicitly optimized on the physical constraints.

[1] Christian Jacobsen, Yilin Zhuang, and Karthik Duraisamy. CoCoGen: Physically-Consistent and Conditioned Score-based Generative Models for Forward and Inverse Problems. pp. 1–25, 2023.

[2] Dule Shu, Zijie Li, and Amir Barati Farimani. A physics-informed diffusion model for high-fidelity flow field reconstruction. Journal of Computational Physics, 478:111972, 2023.

**Weaknesses:**

In their experiment, although their proposed method performs best in satisfying physical constraints, it shows suboptimal performance on other metrics. Moreover, the approach of adding a physical residual term to the loss function is straightforward and offers limited innovation.

**Questions:**

1.	What advantages does the proposed method have in topological optimization task compared to the traditional methods (SIMP)? For optimization problems, it does not seem to be necessary to generate a distribution. The authors mentioned that their method can also provide the displacement fields. Is it a feature that exclusively can be obtained with their method? Could the authors explain in detail?
2.	How does the sampling step in the sample estimation (DDIM) affect the performance? Will the performance improve with more steps?
3.	How much will the scaling c in the loss function affect the performance? Could the authors show the experiment results of different scaling c?

---

> ### Author Response · Authors · 2024-11-24
> **Clarifying model performance and benefit of simple loss**
>
> We thank the reviewer for their valuable feedback and thoughtful questions. Below, we address each point in detail.
>
> **Weaknesses:**
>
> Allow us to first clarify on the metrics: In the Darcy flow example, the PIDM with sample estimate *outperforms all other variants*, including the standard diffusion model and prior SOTA, on the two relevant metrics: residual error and data loss (see Fig. 2). Notably, the PIDM does not only reduce the residual by order of magnitudes but is also more robust to overfitting by matching the underlying physical ground truth.
>
> For the topology optimization study, we fully acknowledge that not all metrics are best for the PIDM. However, the metrics must be interpreted more nuanced: For the compliance and volume fraction error, there will always be a trade-off as adding more material (i.e., increasing the volume error) will always lead to a stiffer structure (i.e., reduced compliance error). Thus, both of these measures have to be taken jointly into account. Depending on what is valued more, the *PIDM performs similarly if not better compared to frameworks specifically designed for this task*, which require auxiliary models and potentially additional 'noised' data [1,2], thus drastically complicating the framework. In contrast, our approach is intuitive and simple to implement, which, while not explicitly listed in the table, underscores the framework’s generality and ease of implementation. This simplicity highlights its potential for broader applicability across various domains.
>
> The reviewer raises the good point that the integration of a residual loss term is 'straightforward', which we in fact fully agree with and would actually like to clarify on this: It was specifically our goal to come up with a method that is *(i) general applicable for any physical residuals, (ii) effective in improving the physical accuracy, and (iii) straightforward to implement*. Alternatives such as [2] with similar objectives require, as mentioned before, auxiliary datasets and models, and we see it as an advantage that our method reduces to a simple extension of the loss. *The fact that we arrive at this simple form is, and we want to emphasize this, the exact intention of our derivations*, which are fully motivated from first principles via virtual observables. In fact, we have shown all relevant approximations and simplifications that must be introduced to exactly arrive at the 'simple' loss given in Eq. 13.
>
> Furthermore, key contributions are the two alternative mean and sample estimates to efficiently evaluate Eq. 13, which is a challenge specific to diffusion models. We would also like to point out that this straightforward integration of physics into this setting was, to the best of our knowledge, not obtained by prior work, though informing diffusion models on physics has been the theme of various contributions, as listed in the manuscript. We suspect that the understanding of the general setting of introducing deterministic constraints into probabilistic diffusion models, as well as specific challenges such as the computationally intensive, iterative sampling, are the prime reasons for this gap in the literature that we fill here.

---

> > ### Author Response · Authors · 2024-11-24
> > **Clarifying relevance for optimization problems and further details**
> >
> > **Question 1:** The distributional perspective is relevant in any scenario in which the solution might not necessarily be unique, which is often the case for optimization problems. For instance, it is well-known in the topology optimization community that for given boundary conditions, many different optimal designs may exist [3]. In these scenarios, generative models are particularly well-suited as they learn the distribution of feasible solutions rather than a deterministic single solution. Equally important, we can easily access these solutions after training by conditional sampling, which can outperform iterative classical methods such as SIMP, which was a prime motivation for prior work applying diffusion models to topology optimization [1,2]. The displacement fields can of course also be predicted with any other diffusion model variant, however, it is only in our setting that we can inform the model of the underlying physical laws of linear elasticity that govern the displacement field. As indicated in Table 1, the physics-informed model predicts displacement fields that are more accurate compared to a purely data-driven model. We have rephrased the sentence to clarify this.
> >
> > **Question 2:** This is a great question that we have investigated in detail. Interestingly, we observed no significant improvements in terms of the residual error reduction with additional steps. We hypothesize that the main reduction of the residual happens in the last steps of the model (where the noise gets minimal) and a 1-step mapping from these small timesteps to the clean sample does not reduce the sample quality notably. We note that there is a significant drawback in adding steps as every additional step notably increases training time. For these two reasons, we did not investigate this approach further.
> >
> > **Question 3:** The scaling $c$ indeed is the critical parameter as it dictates the contribution of data vs physics-driven terms. As $c \to 0$, the model obviously is not meaningfully informed on the residual and converges to the data-driven one. As $c\gg1$, the opposite happens: The data likelihood is ignored and the model aims to minimize a random point on the constraint manifold. The distribution collapses and we effectively recover the classical PINN framework. Since we focus on understanding and clarity, we have shown experimental results of the toy problem of learning a circular distribution, which clearly shows the effect of c for various settings. Essentially, for large $c$, the residual is further reduced but the distribution increasingly collapses to a smaller and smaller domain of the constraint manifold, disregarding the data distribution. At its optimal value, $c$ respects both the data distribution while also improving the constraint fulfillment. We have provided the detailed study in Appendix A.6.1 of the revised manuscript.
> >
> > We would again like to thank the reviewer for their valuable input and hope to have addressed their concerns.
> >
> > [1] François Mazé and Faez Ahmed. Diffusion Models Beat GANs on Topology Optimization. Proceedings of the 37th AAAI Conference on Artificial Intelligence, AAAI 2023, 37:9108–9116, 2023.
> >
> > [2] Giorgio Giannone, Akash Srivastava, Ole Winther, and Faez Ahmed. Aligning Optimization Trajectories with Diffusion Models for Constrained Design Generation. Advances in Neural Information Processing Systems, 2023.
> >
> > [3] Martin P. Bendsøe and Ole Sigmund. Topology Optimization. Springer Berlin Heidelberg, 409 Berlin, Heidelberg, 2004

---

> > > ### Author Response · Authors · 2024-12-02
> > > **Requesting the reviewer for feedback**
> > >
> > > As the discussion period is nearing its end, we kindly ask the reviewer to share their feedback on our rebuttal. We remain available to address any further questions or clarifications regarding the paper or our response.

---

> > > > ### Comment · Reviewer_VMKn · 2024-12-02
> > > > **Raised score**
> > > >
> > > > Thank you for clarifying the strengths of the proposed method and comparsion with other methods in the experiments. I raised the score from 5 to 6.

---

> > > > > ### Author Response · Authors · 2024-12-02
> > > > > **Thanking the reviewer**
> > > > >
> > > > > The authors thank the reviewer for their evaluation and for recognizing the key strengths and advantages of the method.

---

### Official Review · Reviewer_fEo6 · 2024-11-04

**Soundness:** 4
**Presentation:** 3
**Contribution:** 3
**Rating:** 6
**Confidence:** 4

**Summary:**

The paper aims to model random variables using diffusion-based models on data spaces, whose samples follow certain physical constraints described by partial differential equations (PDEs) and boundary conditions. Specifically, these PDE constraints are expressed in the form $\mathcal{F}(x) = 0$ for $x$ in the domain $\Omega$, where $\mathcal{F}$ is a differential operator governing the physical constraints. For example, consider a data space where each data point represents a uniform discretization over a fixed time interval of projectile motion; in this scenario, each data point must comply with the physical laws related to gravity. The paper proposes a regularization objective applied to the generated samples of the diffusion-based models., which does not alter the common training objectives of the models.

To incorporate the physical constraints, the paper introduces a residual term defined as $R(x) = \mathcal{F}(x)$. They model the likelihood of the residual given $x$ as $p(r \mid x) = N(r; R(x), \sigma^2 I)$, where $N$ denotes the normal distribution and $\sigma > 0$ is a positive constant. The training process aims to maximize the likelihood $p(r = 0 \mid x)$, effectively encouraging the residuals to be zero and thus satisfying the physical constraints. With this likelihood function, the paper proposes maximizing the expected log-likelihood of the residuals with respect to the model, in addition to maximizing the evidence lower bound (ELBO) of the diffusion-based models.

Since computing the expected log-likelihood of the residuals with respect to the model is computationally expensive—requiring sampling from the diffusion-based models—the paper proposes sampling $x_0$ for a given $x_t$ instead of generating samples from the prior.

The paper demonstrates the effectiveness of the proposed method through various experiments.

----
Updated the soundness from 3 to 4 and the presentation & contribution from 2 to 3 after the authors' rebuttal.

**Strengths:**

The paper introduces an interesting approach by modeling random variables using diffusion-based models that adhere to physical constraints defined by PDEs and boundary conditions. This integration of physical laws into the modeling process through a regularization objective is particularly compelling. This novel framework has some potential to positively influence the machine-learning community, especially in domains where physical consistency is essential.

The demonstrated effectiveness across various experiments underscores the practicality and impact of the proposed method.

**Weaknesses:**

The paper would benefit from significant improvements in writing clarity.

To fully understand and summarize the contributions, I had to reference several previous studies, which suggests that the paper is not entirely self-contained. For instance, lines 127-129 appear to cover a key concept, yet there is no explanation provided, nor is the term $R$ properly defined.

Additionally, there are ambiguities in the notation—for example, it’s unclear if $x$ in Section 2.1 is intended to represent the same variable as $\xi$ in Equations (5) and (6), raising questions about consistency.

Furthermore, the abstract is overly vague, making it difficult to grasp the paper’s main content and contributions at a glance.

**Questions:**

N/A

---

> ### Author Response · Authors · 2024-11-24
> **Clarifying notation**
>
> We thank the reviewer for acknowledging our 'compelling' contribution that can benefit the machine learning community in the scientific setting.
>
> We also thank them for pointing out difficulties in following the notation introduced in our paper. We have extended the explanations in our revised manuscript (highlighted in red) and clarified our contributions in the abstract. We have also included the additional Appendix A.9 to clarify the assembly of the residual. The variable $\boldsymbol{\xi}$ is always reserved for spatial coordinates that are relevant in the assembly of the residual, while $\boldsymbol{x}_0$ always refers to a data point. This is in our context typically an image that contains the relevant solution field values $\boldsymbol{u}$ at a collection of discretized spatial coordinates $\boldsymbol{\xi}$.
>
> We hope this clarifies potential ambiguities and hope to have addressed the reviewers concerns. If further ambiguities prevail, we are more than happy to address them.

---

> ### Comment · Reviewer_fEo6 · 2024-11-25
>
> Thank you for addressing my concerns in your revised manuscript. I acknowledge the improvements made, particularly in enhancing the clarity of the paper's presentation, including the definition of $\mathcal{R}$. However, since it serves the essential role of the proposed method, I believe the definition should move to the main manuscript. I appreciate your efforts and am pleased to reflect these revisions in my updated evaluation.

---

> ### Author Response · Authors · 2024-12-02
> **Thanking the reviewer**
>
> We thank the reviewer for their constructive feedback and for acknowledging the improvements in our revised manuscript. We appreciate their recognition of our contributions and the updated evaluation. Upon acceptance, we will move the definition of $\mathcal{R}$ to the main article.

---

### Official Review · Reviewer_szYR · 2024-11-04

**Soundness:** 3
**Presentation:** 2
**Contribution:** 3
**Rating:** 6
**Confidence:** 4

**Summary:**

The paper proposes a method which introduces a virtual observable (Rixner & Koutsourelakis, 2021) to the score-matching denoising diffusion training objective (Song et al., 2021).
The authors provide a theoretical argument (see Appendix A.1) showing that, in ideal settings, this enables the diffusion model to recover the true distribution while enforcing physical constraints.
Additionally, the authors theoretically motivate a simplification of the augmented objective (see Appendix A.3), ignoring the likelihood ratio, and justify this by assuming that ongoing optimization will reduce the introduced bias.
The authors present experiments on fluid flow (Darcy flow) and structural optimization (topology optimization).
These experiments indicate improved reduction in PDE residual compared to state-of-the-art methods and provide evidence that the additional training objective does not compromise data likelihood and may instead serve as an effective regularizer against overfitting.

**Strengths:**

1. Originality: The paper incorporates previous work on virtual observables (Rixner & Koutsourelakis, 2021) to train denoising diffusion models with embedded physical constraints. In the realm of physics-informed diffusion models, this approach contrasts with methods like PG-Diffusion (Shu et al., 2023) and CoCoGen (Jacobsen et al., 2023), which incorporate physics via auxiliary models or adjustments post-training.
2. Quality: The paper provides coherent theoretical arguments for the application of virtual observables in diffusion models (see Appendix A.1 and A.3) and implements an experimental setup on benchmark tasks, following configurations from established related work (see Appendix A.6.2 and A.6.3).
3. Clarity: The paper is clearly structured and well-written.
4. Significance: The method could enhance generative modeling for scientific applications by producing physically consistent samples, as shown in fluid dynamics and structural optimization benchmarks.

**Weaknesses:**

1. Clarity: The approach is substantially grounded in prior work on virtual observables (Rixner & Koutsourelakis, 2021), making the contribution more incremental than suggested.
Greater clarity would be achieved by explicitly framing the contribution as an adaptation of this framework tailored for denoising diffusion models.

**Questions:**

The augmentation of the denoising diffusion model objective to incorporate virtual residuals represents a noteworthy contribution to the field. However, it would enhance clarity and rigor if the authors explicitly positioned this adaptation as their primary theoretical contribution.

---

> ### Author Response · Authors · 2024-11-24
> **Clarifying distinction from Rixner & Koutsourelakis, 2021**
>
> We thank the reviewer for their thoughtful review and feedback. We sincerely appreciate their recognition of our paper’s originality, quality, clarity, and significance, as well as the potential of our method to enhance generative modeling for scientific applications.
>
> We understand the concern regarding the clarity of our contribution, particularly in relation to prior work on virtual observables [1], which we have duly acknowledged in our manuscript. As a starting point, we adopt the virtual observable concept as presented in Eq. 8 of [1] (corresponding to Eq. 7 in our manuscript). *However, the remainder of our manuscript differs significantly in all subsequent derivations (and implementation).*
>
> Unlike [1], we do not compute an ELBO approximation over the joint distribution. Instead, we treat the physical likelihood as an auxiliary training objective, analogous to the original framework of PINNs [2]. For this auxiliary physical likelihood, we avoid approximating the ELBO and instead evaluate it exactly when sampling from the model. The theoretical insights underlying this auxiliary loss term, including its exact sampling mechanism, are detailed in Appendix A.1 and A.3.
>
> The key challenge in the context of diffusion models lies in the computational infeasibility of exact sampling, requiring accurate and efficient alternatives. Addressing this challenge forms a major contribution of our work, which is completely irrelevant in [1]. Specifically, we propose two novel approaches for efficient sampling—both surpassing prior state-of-the-art methods in terms of residual error that may be of independent interest:
>
> **Mean Estimation**:
> This method leverages the estimated mean of the clean signal given a noisy input, avoiding additional forward passes. However, it introduces an inconsistency due to the Jensen gap, as the residual evaluated on the mean does not necessarily equal the mean of the residuals.
>
> **Sample Estimation**:
> In this approach, we obtain a sample estimate of the clean signal through accelerated sampling schemes, particularly deterministic Denoising Diffusion Implicit Models (DDIM) [3]. While requiring an additional forward pass, this method significantly reduces inconsistencies between the data and residual losses. To the best of our knowledge, this approach has not previously been integrated into the training process (as opposed to inference). We provide theoretical justifications for the superior performance of sample estimation over mean estimation in our context, supported by clear numerical evidence (see Figure 2 in the manuscript).
>
> To summarize -- while we also start with the virtual observable as introduced in Eq. 8 of [1], our subsequent setting and contributions are fundamentally distinct, particularly in addressing the unique challenges of diffusion models. Unlike prior work, we do offer a fully first-principle based derivation of the integration of physics into diffusion models, do highlight all relevant simplifications and potentially introduced biases, and demonstrate SOTA performance on most relevant metrics in two relevant settings.
>
> We hope these clarifications address the reviewer's concerns and highlights the originality and significance of our work and would appreciate a reconsideration of the score.
>
> [1] Maximilian Rixner and Phaedon Stelios Koutsourelakis. A probabilistic generative model for semisupervised training of coarse-grained surrogates and enforcing physical constraints through virtual observables. Journal of Computational Physics, 434, 2021.
>
> [2] M. Raissi, P. Perdikaris, and G.E. Karniadakis. Physics-informed neural networks: A deep learning
> framework for solving forward and inverse problems involving nonlinear partial differential
> equations. Journal of Computational Physics, 378:686–707, 2019.
>
> [3] Jiaming Song, Chenlin Meng, and Stefano Ermon. Denoising Diffusion Implicit Models. International Conference on Learning Representations, 2021.

---

> > ### Comment · Reviewer_szYR · 2024-11-25
> > **Thank you for the clarification**
> >
> > Thank you for providing detailed clarifications regarding your contributions.
> > Especially the distinction from prior work and the novel approaches for sampling in diffusion models were insightful.
> > Based on this response, I decided to raise my score from 5 to 6.

---

> > > ### Author Response · Authors · 2024-12-02
> > > **Thanking the reviewer**
> > >
> > > We thank the reviewer for their thoughtful feedback and for recognizing our contributions. We greatly appreciate their engagement and the decision to raise the score.

---

### Author Response · Authors · 2024-12-02
**Final comment**

We thank all reviewers for their thorough evaluation and constructive feedback. Their comments have helped us better articulate our key contributions and strengthen our manuscript. Throughout the discussion period, we have clarified the distinct aspects of our framework: operating in a data-driven regime while improving physical consistency, introducing efficient mean and sample estimation techniques for training, and achieving state-of-the-art performance in residual reduction with a straightforward implementation. We are encouraged that all reviewers have raised their scores after our responses, acknowledging both the originality and significance of our work for scientific machine learning applications.

If accepted, we commit to incorporating all the suggested improvements in the camera-ready version, including additional references and clearer notation.

Sincerely,

The Authors

---

### Meta-Review · Area_Chair_kTjp · 2024-12-22

**Metareview:**

The authors proposed a novel strategy for generating physical data under physical constraints by incorporating the training objective of diffusion models and physical PDEs directly into the loss function. The data required for PDEs are estimated using either one-step score-based mean estimation or two-step DDIM sampling. The authors evaluated their approach against other physics-guided machine learning models on tasks such as Darcy flow generation and topology optimization. The reviewers had some concerns but all were addressed in the rebuttal. One remaining concern is the small scale of the experiments; however, I believe this is outweighed by the contributions of this paper. Therefore, I recommend it for acceptance.

**Additional Comments On Reviewer Discussion:**

Partially due to the poor writing clarity of the original paper, the reviewers had some concerns on some technical details. Fortunately, all these concerns were addressed by the authors, and all the reviewers raised their scores after rebuttal and discussion. Reviewer oHnR still had one concern on the scale of the experiments and gave 5 finally. I personally think the experiments in the paper have a reasonable scale, and the contribution is sufficient for acceptance.

---

### Decision · Program_Chairs · 2025-01-22

Accept (Poster)